# Diversity-Enhanced Reasoning for Subjective Questions

**Yumeng Wang**[1*]  **Zhiyuan Fan**[1*]  **Jiayu Liu**[1*]  **Jen-tse Huang**[3]  **Yi R. (May) Fung**[1,2†]
[1]Hong Kong University of Science and Technology  [2]MMSense Lab  [3]Johns Hopkins University
ywanglu@connect.ust.hk    yrfung@cse.ust.hk

## Abstract

Large Reasoning Models (LRMs) with long chain-of-thought capabilities, optimized via reinforcement learning with verifiable rewards (RLVR), excel at **objective reasoning** tasks like mathematical problem solving and code generation. However, RLVR is known for degrading generation diversity, which causes LRMs to fall short on **subjective reasoning** that has multiple answers depending on different role perspectives. While recent studies recognize the importance of diversity-enhanced training in objective reasoning, limited attention has been given to subjective tasks. In this paper, we find that subjective reasoning can be improved by introducing perspective diversity and token-level diversity, with the former one providing a coherent scaffolding anchored to a real-world stakeholder group and the latter one broadening the answer search space. We propose **MultiRole-R1**, a diversity-enhanced training framework featuring an unsupervised data construction pipeline that synthesizes reasoning chains incorporating various role perspectives. It also employs reinforcement learning via Group Relative Policy Optimization with reward shaping, taking diversity as a reward signal in addition to verifiable reward. Training on subjective tasks solely, MultiRole-R1 increases the in-domain and out-of-domain accuracy by 14.1% and 7.64%, and even enhances the performance on advanced math reasoning such as AIME 2024. We further show that diversity is a more consistent indicator of accuracy than reasoning length. The code and data of this work is available at `https://github.com/yumeng-10/multirole-r1`.

## 1 Introduction

Advances in DeepSeek-R1 (DeepSeek-AI et al., 2025) and OpenAI o1-style (Jaech et al., 2024) models with long Chain-of-Thoughts (CoT) capabilities (Wei et al., 2023) have substantially improved performance on challenging reasoning tasks, particularly in objective domains such as commonsense (Talmor et al., 2019) and mathematical reasoning (Yu et al., 2025; Wu et al., 2024b; Cobbe et al., 2021; Wang et al., 2025a; Guo et al., 2025, *inter alia*).

Notably, this type of model is trained via reinforcement learning with verifiable rewards (RLVR), which induces a diversity degradation in the model generation (Song et al., 2025b; Dang et al., 2025; Zhao et al., 2025; Wu et al., 2025). This greatly undermines the real-world application, since diversity is crucial for effective sampling for test-time scaling (Yue et al., 2025). Recent studies offer several solutions to enhance diversity in RL training (Song et al., 2025a; Yan et al., 2025), but they mostly focus on objective reasoning.

In contrast to objective tasks, subjective questions are fundamentally different from objective questions since there are no definitive right or wrong answers (Khurana et al., 2024; van der Meer et al., 2024; Wang et al., 2025b; Jentzsch & Kersting, 2023; Wu et al., 2024a): the responses can vary greatly depending on the role or stakeholder perspective.

This challenge cannot be solved by current diversity-enhanced training approaches in objective domain since they rely on a single ground truth in optimization. This design inherently trains the model to find one correct answer, making it incapable of generating reasoning that arrives at

---

[*]Equal contribution, with Zhiyuan Fan as the student project leader (see Author Contributions section).
[†]Corresponding author.

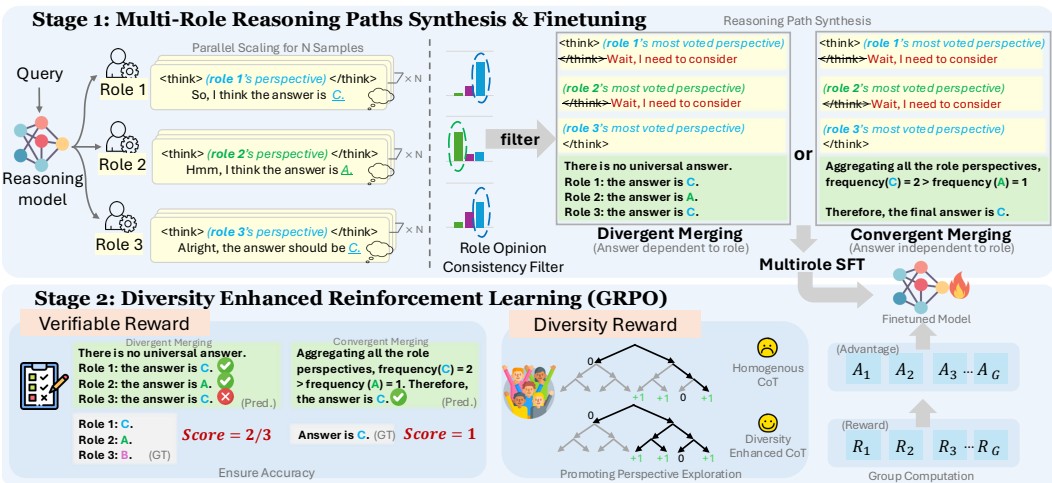

Figure 1: Illustration of MultiRole-R1 framework. **Stage 1** (enhance perspective diversity): LRMs generate seed roles with contrastive opinions, and sample diverse reasoning paths from different roles. We concatenate paths from different perspectives into a CoT, and then finetune the model to follow the multi-role reasoning format. **Stage 2** (enhance token-level diversity): we utilize GRPO with diversity reward shaping. Verifiable rewards are applied depending on whether the ground-truth varies by roles. We take diversity as an additional reward to promote exploration efficiency.

the multiple valid outcomes required by subjective questions. To tackle this, existing research on subjective reasoning mainly falls in two categories: multi-agent debate (Aoyagui et al., 2025; Cheng et al., 2024; Liu et al., 2025c) and prompting-based methods (Wang et al., 2024b; Lv et al., 2024), with no training methods specifically designed for subjective questions.

In this paper, we address this challenge by proposing **MultiRole-R1**, a diversity-enhanced RL training framework to improve LLM subjective reasoning. Specifically, we incorporate two levels of diversity: (1) *Semantic-level diversity* (or *perspective diversity*), which trains the model to incorporate multiple relevant real-world stakeholder perspectives; (2) *Token-level diversity*, which broadens the search space of the reasoning chains. In particular, we argue that role perspective diversity is key to this challenge: instead of just seeking random variation, roles provide coherent scaffolding that ensures the diverse outputs are semantically relevant and anchored to real-world groups and stakeholders' viewpoints (Xu et al., 2025; Wang et al., 2025b).

We conduct a pilot analysis to determine the optimal number of roles and the reasoning length of the generated paths. MultiRole-R1 subsequently finetunes the model on self-synthesized reasoning paths to enhance *semantic-level diversity*, instructing it to self-teach on multi-role generation and role reasoning, as shown in Figure 1. Furthermore, to enhance the *token-level diversity*, MultiRole-R1 applies a diversity reward function combining an array of existing token-level diversity metrics, such as lexical diversity, structural diversity, and discourse diversity. This is used as a reward signal in addition to the verifiable reward in Generalized Reward Policy Optimization (GRPO).

To evaluate the effectiveness and generalizability of our approach, we train DeepSeek-R1 series models and Qwen-3-8B using MultiRole-R1 and test them on both subjective and objective questions. Results show that MultiRole-R1 boosts performance by an average of 14.1% on three in-domain (ID) subjective tasks, and 7.64% on four out-of-domain (OOD) tasks that include both subjective and objective questions. Interestingly, our approach even achieves a performance gain on the OOD advanced math reasoning dataset AIME 2024 by 5.78%. Our further analysis shows that among the 10.6% average performance gain of MultiRole-R1, 8.3% is contributed by multi-role SFT, and 6.6% is contributed by GRPO with diversity reward shaping. This verifies the necessity of incorporating perspective diversity in subjective reasoning, and also corroborates the cruciality of token-level diversity in test-time scaling. Moreover, we find a strong per-task correlation between diversity and accuracy ($r = 0.74$), which markedly outweighs the correlation between length and accuracy ($r = 0.55$). This result extends the previous finding of a correlation between diversity and task performance in objective tasks to subjective tasks, indicating that diversity is a more consistent indicator of accuracy than reasoning length. Our contributions can be summarized as follows:

- To our knowledge, we are the first to introduce diversity-enhanced training for subjective reasoning tasks. We propose MultiRole-R1, a training paradigm that incorporates unsupervised reasoning path synthesis and GRPO with diversity reward shaping, which effectively enables LRM to include diverse perspectives and generate multiple different answers in subjective reasoning.

- We verify MultiRole-R1 on four models using solely subjective questions for training. Results show that the models achieve state-of-the-art performance in three ID and four OOD tasks, and can generalize to advanced math reasoning such as AIME 2024.

- Our analysis highlights diversity as a more consistent indicator of accuracy than reasoning length.

## 2 PILOT ANALYSIS

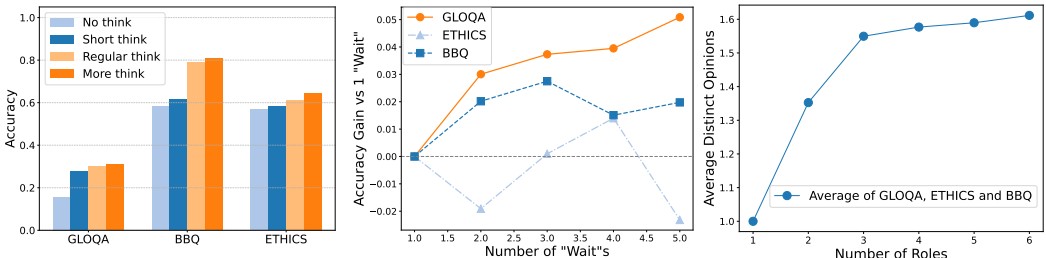

Figure 2: *(a)* The performance of Deepseek-R1-Distill-Qwen-7B (DeepSeek-AI et al., 2025) under different reasoning length settings across different datasets. The bar chart shows that longer reasoning chains result in higher accuracy on subjective tasks. *(b)* Accuracy gain of Deepseek-R1-Distill-Qwen-7B (DeepSeek-AI et al., 2025) when trailing with more wait tokens. *(c)* Demonstration of the number of distinct opinions increases as more roles are involved in a single reasoning chain.

Inspired by previous work (Zelikman et al., 2022; Peng et al., 2025; Liu et al., 2026) that fine-tuning on self-synthesized CoT improves reasoning, we extend this methodology by enabling the model to self-improve on synthesizing diverse perspectives into a single reasoning chain. To achieve this, we concatenate multiple single-role reasoning chains into one long reasoning path, as shown in Figure 1 Stage 1. One natural question is how to decide the format of the multi-role reasoning chain: how many role perspectives to include, and how long should the reasoning path be? On the one hand, we want to incorporate more roles to cover comprehensive perspectives and leverage the self-correction ability of the long reasoning chain. On the other hand, we want to control the reasoning length within an optimal range, as excessive verbosity leads to performance degradation (Hassid et al., 2025) and is computationally expensive.

To answer the question, our pilot analysis aims to study the optimal number of roles and reasoning length. We primarily consider Deepseek-R1-Distill-Qwen-7B, and a set of subjective questions sampled from GlobalOpinion QA (DURMUS et al., 2024) and BBQ (Parrish et al., 2022) datasets. To elicit multiple viewpoints in a single reasoning chain, we employed budget-forcing (Muennighoff et al., 2025), which replaces the end-of-thinking token with a continuation token (i.e., "wait, I need to think from {role}'s perspective"), to divert the model to a different role perspective. These roles, designed to be mutually contrasting in opinions, were pre-generated by the base model via prompting.

Besides budget-forcing that produces longer paths than the regular reasoning, we also compare it with settings that are shorter than regular reasoning, with examples in Appendix C.2. We observe that reasoning lengths longer than regular think (i.e., more think) significantly outperform other settings, as shown in Figure 2 (a). This motivates us to focus on more think setting, which is about how many wait tokens should be appended. The following highlights the main findings of the pilot analysis:

**Scaling Law of Reasoning Length** As shown in Figure 2 (b), increasing the number of "Wait" tokens generally leads to performance improvements across most tasks, where the gains mostly peak around **three** "Wait" and diminish or even degrade beyond that point.

**Scaling Law of Role Perspectives** Since our tasks are in the format of multiple choice answers, the number of opinions can be simply counted as the number of choices. Results in Figure 2 (c) show that the number of distinct opinions increases as more roles are involved, with number of roles $n = 3$ as a turning point where the increase of roles provides less salient information gain compared to the previous. Hence, we incorporate three roles in the path generation.

## 3 METHODOLOGY

As illustrated in Figure 1, our framework consists of two stages: multi-role reasoning paths synthesis & finetuning and diversity-enhanced reinforcement learning. Formally, given an input subjective question $\mathcal{Q}$ and a reasoning model $\mathcal{M}$, our goal is to diversify the reasoning path $\mathcal{T}$.

### 3.1 MULTI-ROLE REASONING PATHS SYNTHESIS & FINETUNING

The objective of this stage is to enhance perspective diversity: besides training the model to "think deeper from its perspective" (DeepSeek-AI et al., 2025; Jaech et al., 2024), we also train the model to consider "from which perspective to think".

**Multi-Role Exploration and Sampling**     To model multi-perspective reasoning, we first identify $n$ context-relevant roles (e.g., domain experts, stakeholders, or personas) through few-shot prompting (Brown et al., 2020; Liu et al., 2025a), denoted as $\mathcal{R} = \{\mathcal{R}_1, \mathcal{R}_2, ..., \mathcal{R}_n\}$. In particular, we prompt the model to generate roles with conflicting viewpoints. The motivation for this is to explore diverse available perspectives. Given a candidate role $\mathcal{R}_i$, LLM $\mathcal{M}$ and question $\mathcal{Q}$, we define the selection probabilities:

$$P(\mathcal{R}_i|\mathcal{Q}) = softmax(\mathbb{E}[\mathcal{M}(\mathcal{R}_i|\mathcal{Q})] + \alpha\mathbb{E}_{\mathcal{R}_i}[1 - \text{sim}(\mathcal{R}_i, \mathcal{R}_j)]), \tag{1}$$

where $sim(\mathcal{R}_i, \mathcal{R}_j) = cos(\mathbf{h}_{\mathcal{R}_i}, \mathbf{h}_{\mathcal{R}_j}|\mathcal{Q})$ and $\mathbf{h}$ denotes the LLM embedding. The intuition for this is to prioritize role answers that are relevant to the question and contrastive to the existing opinions.

**Self-Consistency Filtering**     For each role $\mathcal{R}_i$, we sample $k$ reasoning paths from the decoder with temperature $\tau = 1$, denoted as $\mathcal{M}(Q, \mathcal{R}_i) = \mathcal{T}_{\mathcal{R}_i} = \{\mathcal{T}_{\mathcal{R}_i}^{(1)}, \mathcal{T}_{\mathcal{R}_i}^{(2)}, ..., \mathcal{T}_{\mathcal{R}_i}^{(k)}\}$. To ensure the coherence among different responses of each role, we then apply self-consistency filtering (Chen et al., 2023; Wang et al., 2025c) through majority voting and only keep the most consistent answer:

$$\hat{\mathcal{T}}_{\mathcal{R}_i} = argmax \sum_{j=1, \mathcal{T} \in \mathcal{T}_{\mathcal{R}_i}}^{k} \mathbb{1}(\mathcal{T} \equiv \mathcal{T}_{\mathcal{R}_i}^{(j)}), \tag{2}$$

where $\mathbb{1}$ is the indicator function and $\equiv$ denotes semantic equivalence (e.g. same roles give different answers). This approach extends ensemble methods by decoupling role-specific reasoning trajectories, ensuring that conflicting viewpoints remain independently generated and self-consistent.

**Reasoning Structure Generation**     Given $m$ filtered role perspectives $\{\hat{\mathcal{T}}_{\mathcal{R}_1}, ..., \hat{\mathcal{T}}_{\mathcal{R}_m}\}$, we generate random combinations of role orderings $\Pi$ to avoid the effect of position bias (Zheng et al., 2023a). For example, given a multi-role combination $\pi = \{\mathcal{R}_i, \mathcal{R}_j, \mathcal{R}_k\}$, we construct the training data as:

$$\mathcal{D}_{\text{train}} = \bigcup_{\pi \in \Pi} \{(\mathcal{Q} \oplus \hat{\mathcal{T}}_{\mathcal{R}_i} \oplus \hat{\mathcal{T}}_{\mathcal{R}_j} \oplus \hat{\mathcal{T}}_{\mathcal{R}_k}) \mid \pi\}. \tag{3}$$

We consider two merging strategies depending on task type to allow dynamic integration of role reasoning paths: (1) divergent merging: for tasks where roles are expected to provide different answers, the final prediction is derived through a weighted aggregation of the various viewpoints; (2) convergent merging: for tasks where roles should yield a consistent answer, a consensus is reached via majority voting within the reasoning sequence.

**Multi-Role Supervised Finetuning**     To ensure data quality, we apply both rule-based and automatic filtering strategies to the merged data. To mitigate verbosity bias (Zheng et al., 2023b) and reasoning shortcut behavior, we remove the top and bottom 10th percentiles of responses by length. We also discard instances with formatting errors or invalid string patterns. This filtering process yielded a final training set of 2,700 entries, with detailed decomposition shown in Table 8. For comparison against our self-consistency filtering method, we also applied a supervised ground-truth filtering approach. For the supervised approach, roles are sampled from the built-in role pool of the ground truth data, and we only keep the trajectories where roles reasoning are correct. Comparison results of two filtering strategies are in Section 5.

## 3.2 Diversity Enhanced Reinforcement Learning

This stage aims to enhance the diversity of the reasoning chain, broadening the answer search space. We adopt Group Relative Policy Optimization (GRPO) (Shao et al., 2024) for Multi-Role reinforcement learning, which is trained on top of the SFT model. GRPO optimizes the policy by sampling a group of candidate outputs for each prompt and comparing their reward. We incorporate two types of rewards: (1) a *multirole-aware* verifiable reward $\mathbf{R}_{acc}$, provided by a verifiable reward model that checks role-based reasoning answer correctness, and (2) a diversity reward $\mathbf{R}_{div}$ computed from the input text as a shaping signal. The total shaped reward is formulated by $\mathbf{R} = \delta\mathbf{R}_{acc} + (1 - \delta)\mathbf{R}_{div}$. Note that the computation of $\mathbf{R}_{acc}$ and $\mathbf{R}_{div}$ are consistent with the definition of accuracy and diversity in Section 4.4.

This follows the reward-shaping paradigm (Ng et al., 1999), where the auxiliary $\mathbf{R}_{div}$ guides learning without changing the optimal policy. Detailed setting is presented in Appendix B.

During training, we observe a synergetic effect of optimizing the diversity and accuracy objectives. This also mitigates issues observed in the SFT baseline, such as excessive verbosity and repetitive reasoning (Toshniwal et al., 2025). Finally, note that GRPO computes group (G) advantages $\mathbf{A}_1, \mathbf{A}_2, \ldots, \mathbf{A}_G$ instead of standard reward, which is given by: $\mathbf{A}_i = (\mathbf{R}_{i,t} - \mu)/\sigma, t \in \{1, \ldots, |G|\}$. Hence, a group with uniform rewards (all 0s or all 1s) would give zero advantage and stall learning. By adding the diversity term, we ensure intra-group reward variance, enabling informative gradients and continued optimization. Mathematical proofs and detailed derivations are provided in Appendix A.

## 4 Experiment and Results

### 4.1 Datasets

We **train** our model on 3 subjective tasks: ambiguous question answering (BBQ) by Parrish et al. (2022), opinion-based QA (GlobalOpinionQA) by DURMUS et al. (2024), and ethical dilemma (ETHICS) by Hendrycks et al. (2021). To evaluate the effectiveness and generalizability of our approach, we **test** on 4 additional datasets: cultural natural language inference (CALI) by Huang & Yang (2023), commonsense reasoning (CSQA) by Talmor et al. (2019), and mathematical reasoning (GSM8K) by Cobbe et al. (2021). We also evaluate our method on the more advanced math reasoning task AIME 2024[1] and present results in Section 5. Specifically, for the out-of-domain (OOD) data, CALI consists of subjective questions, while CSQA, GSM8K and AIME 2024 consist of objective questions. It is worth noting that among these benchmarks, GLOQA and CALI have role-dependent ground truths, whereas the others rely on a single ground truth for all roles.

### 4.2 Baselines

**In-Context Learning** We first incorporate the following in-context learning (Brown et al., 2020) settings: (1) Zero-Shot CoT (Kojima et al., 2023; Wei et al., 2023), (2) Role Playing Prompting (Kong et al., 2024) and (3) Self-Refine Prompting (Madaan et al., 2023).

**More Think** As observed by Muennighoff et al. (2025), extending the reasoning chain length can further enhance the reasoning capabilities of o1-style models. In MultiRole-R1, this is achieved by suppressing the end-of-thinking token and appending a continuation string (e.g., "wait, I need to think from {role}'s perspective") to encourage extended reasoning from a different role perspective. In the *more think* baseline, we employ a reasoning length three times longer than *regular think*, as it offers a balance between efficiency and accuracy based on our pilot analysis.

**Supervised Finetuning** We perform supervised finetuning on the base model on the self-consistency filtered dataset of size 2,700. Our SFT training and evaluation are conducted via Llama-Factory (Zheng et al., 2024).

**Direct Preference Optimization** We introduce another SFT+RL pipeline as an comparison to MultiRole-R1. In this setting, DPO is applied to the self-consistent SFT model, using ground-

---

[1]https://huggingface.co/datasets/Maxwell-Jia/AIME_2024

Table 1: Main results of the baselines (specified in Section 4.2) and our proposed method. *Acc.* is the pass@1 accuracy of the task (in %) and *Div.* measures the length normalized diversity score of the reasoning chain (in %). We include two ablations of MultiRole-R1, including SFT on self-consistency filtered data only (Ours *SelfConsis SFT*), and also SFT with vanilla GRPO (Ours *SelfConsis SFT + GRPO*). "GRPO(RS)" represents GRPO with reward shaping, which is used in MultiRole-R1. OOD denotes the datasets that are for testing only.

| Model | BBQ | | GLOQA | | ETHICS | | CALI (OOD) | | CSQA (OOD) | | GSM8K (OOD) | |
|---|---|---|---|---|---|---|---|---|---|---|---|---|
| | Acc. | Div. | Acc. | Div. | Acc. | Div. | Acc. | Div. | Acc. | Div. | Acc. | Div. |
| *(R1-Distill-Qwen-7B)* | | | | | | | | | | | | |
| Zero-shot CoT | 62.45 | 56.02 | 32.62 | 65.88 | 51.82 | 36.14 | 50.30 | 52.22 | 63.06 | 83.83 | 80.48 | 68.08 |
| Self-Refine | 74.08 | 73.13 | 43.13 | 59.88 | 52.19 | 37.36 | 50.76 | 66.09 | 54.02 | 77.61 | 87.01 | 80.37 |
| Role-Play | 73.61 | 74.68 | 41.67 | 77.75 | 50.83 | 37.89 | 52.69 | 67.43 | 55.07 | 76.20 | 85.66 | 72.87 |
| *More think* | 80.76 | 80.44 | 36.42 | 86.90 | 64.44 | 81.53 | 60.45 | 78.82 | 64.50 | 85.85 | 82.05 | 81.79 |
| *SelfConsis SFT* | 85.88 | 81.67 | 43.13 | 85.58 | 67.45 | 82.19 | 67.35 | 78.94 | 66.88 | 83.10 | 80.62 | 74.87 |
| *SelfConsis SFT+DPO* | 86.41 | 60.43 | 44.20 | 61.17 | 67.28 | 68.51 | 68.19 | 64.09 | 67.24 | 69.83 | 81.51 | 67.56 |
| *SelfConsis SFT+GRPO* | 94.30 | 85.52 | 47.22 | 87.46 | 69.50 | 85.40 | 70.83 | 82.15 | 69.43 | 86.85 | 85.58 | 82.16 |
| **MultiRole-R1** *SelfConsis SFT+GRPO(RS)* | **94.50** | **86.25** | **49.10** | **89.67** | 66.83 | **87.27** | **70.85** | **83.31** | 66.94 | **87.96** | **87.36** | **82.46** |
| *(R1-Distill-Llama-8B)* | | | | | | | | | | | | |
| Zero-shot CoT | 80.89 | 79.92 | 38.41 | 87.07 | 62.46 | 79.44 | 60.84 | 73.98 | 67.21 | 83.39 | 78.87 | 76.52 |
| Self-Refine | 74.20 | 75.85 | 43.19 | 81.11 | 60.96 | 80.17 | 61.95 | 78.87 | 63.77 | 82.61 | 80.95 | 81.24 |
| Role-Play | 74.40 | 80.91 | 44.87 | 83.02 | 64.24 | 79.78 | 62.70 | 77.32 | 67.32 | 82.27 | 77.33 | 75.02 |
| *More think* | 88.20 | 84.11 | 44.04 | 87.19 | 68.06 | 83.99 | 64.41 | 80.30 | 70.42 | 84.73 | 83.30 | 84.12 |
| *SelfConsis SFT* | 89.69 | 82.64 | 48.17 | 87.26 | 70.56 | 81.36 | 70.05 | 79.77 | 70.86 | 83.88 | 86.02 | 81.53 |
| *SelfConsis SFT+DPO* | 90.52 | 80.43 | 48.89 | 80.97 | 71.22 | 83.64 | 69.81 | 76.62 | 71.28 | 82.71 | 86.34 | 81.81 |
| *SelfConsis SFT+GRPO* | 94.47 | 85.75 | 48.55 | 89.36 | 75.63 | 87.89 | 69.26 | 83.37 | 73.71 | 87.96 | 87.49 | 85.31 |
| **MultiRole-R1** *SelfConsis SFT+GRPO(RS)* | **95.55** | **89.58** | **49.06** | **91.78** | **75.84** | **96.54** | **71.48** | **90.55** | **75.12** | **92.98** | **89.79** | **88.45** |
| *(R1-Distill-Qwen-14B)* | | | | | | | | | | | | |
| Zero-shot CoT | 85.01 | 68.06 | 36.82 | 79.18 | 73.63 | 72.20 | 75.05 | 71.83 | 75.85 | 83.09 | 85.58 | 70.68 |
| Self-Refine | 90.42 | 80.13 | 49.04 | 69.40 | 76.48 | 83.22 | 71.28 | 78.22 | 76.55 | 82.31 | 84.73 | 80.24 |
| Role-Play | 91.18 | 81.87 | 49.90 | 75.73 | 77.16 | 75.30 | 67.41 | 70.74 | 75.71 | 79.05 | 91.50 | 76.60 |
| *More think* | 94.57 | 80.67 | 41.60 | 84.04 | 79.36 | 83.33 | 75.90 | 76.81 | 79.36 | 81.77 | 88.76 | 80.94 |
| *SelfConsis SFT* | 94.40 | 75.06 | 50.98 | 81.04 | 81.45 | 71.34 | 76.08 | 73.65 | 81.50 | 77.60 | 91.61 | **91.62** |
| *SelfConsis SFT+DPO* | 94.98 | 67.21 | 51.33 | 65.03 | 81.88 | 42.32 | 75.82 | 71.71 | 79.82 | 69.41 | 90.92 | 68.69 |
| *SelfConsis SFT+GRPO* | 95.98 | 86.88 | 51.73 | 90.33 | 83.50 | 89.42 | 75.65 | 84.92 | 81.19 | 89.64 | 91.87 | 86.36 |
| **MultiRole-R1** *SelfConsis SFT+GRPO(RS)* | **97.50** | **90.17** | **53.98** | **91.32** | **86.00** | **92.89** | **76.50** | **89.08** | **82.00** | **91.61** | **93.43** | 87.24 |
| *(Qwen3-8B)* | | | | | | | | | | | | |
| Zero-shot CoT | 91.71 | 70.10 | 42.13 | 60.99 | 72.29 | 76.68 | 73.40 | 57.43 | 80.81 | 57.84 | 85.41 | 70.49 |
| Self-Refine | 88.93 | 53.07 | 45.25 | 85.00 | 70.64 | 49.60 | 69.40 | 48.16 | 69.22 | 50.99 | 84.91 | 82.31 |
| Role-Play | 89.77 | 41.58 | 47.57 | 54.89 | 70.67 | 47.83 | 70.26 | 50.49 | 72.98 | 48.17 | 93.58 | 81.93 |
| *More think* | 95.18 | 74.20 | 43.39 | 78.98 | 78.26 | 72.45 | 75.10 | 68.44 | 81.20 | 73.07 | 90.02 | 75.07 |
| *SelfConsis SFT* | 94.05 | 74.02 | 50.32 | 77.07 | 78.39 | 68.35 | 75.96 | 72.78 | 81.00 | 73.50 | 91.62 | 70.23 |
| *SelfConsis SFT+DPO* | 94.21 | 74.83 | 50.68 | 74.14 | 78.09 | 50.31 | 76.10 | 69.58 | 80.45 | 64.87 | 91.84 | 65.32 |
| *SelfConsis SFT+GRPO* | 95.91 | 85.47 | 51.37 | 87.74 | 79.82 | 86.84 | 77.83 | 80.36 | 81.19 | 86.40 | 91.97 | 85.98 |
| **MultiRole-R1** *SelfConsis SFT+GRPO(RS)* | **96.98** | **88.15** | **51.72** | **89.88** | **81.95** | **89.09** | **77.95** | **83.84** | **82.10** | **87.93** | **94.98** | **86.93** |

truth-hinted role answers as positive samples and inconsistent role answers as negative samples.

## 4.3 MODELS

Our experiment is performed on DeepSeek-R1 series (DeepSeek-AI et al., 2025) including R1-Distill-Qwen-7B, R1-Distill-Llama-8B and R1-Distill-Qwen-14B models, and Qwen3-8B (Qwen Team, 2025) with reasoning mode.

## 4.4 METRICS

**Accuracy** Taking into account the subjective nature of role-based reasoning where the ground truth for subjective questions vary across different roles, we adopt two different perspective merging strategies during evaluation. The ground truth of the dataset $\mathcal{G}$ is defined as a role ($r$) ground-truth ($g$) pair, defined as $\mathcal{G} = \{r_i : g_i\}_{i=1}^{|\mathcal{G}|}$. If not specified, accuracy refers to pass@1 accuracy.

**(1) Divergent Merging:** for tasks such as CALI and GLOQA, each role $i$'s answer $a_i$ is compared with the corresponding ground truth $g_i$, where the divergent accuracy is given by: $Acc_{div} = \frac{1}{n} \sum_{i=1}^{n} \mathbb{1}[a_i = g_i]$.

**(2) Convergent Merging:** datasets like BBQ, ETHICS, CSQA, GSM8K and AIME 2024 have answers invariant with role-perspectives. We aggregate different role's answer to obtain a consensus, and then compare it to the ground truth: $\hat{a} = argmax \sum_i \mathbb{1}(a_i = \hat{a}), Acc_{con} = \frac{1}{n} \sum_{i=1}^{n} \mathbb{1}[\hat{a}_i = g_i]$.

**Diversity** To quantify the diversity of model-generated reasoning, we design a composite metric that captures multiple level of linguistic diversity, including lexical, structural and discourse domains. Inspired by prior work on lexical and entropy-based diversity in natural language generation (Li et al., 2016; Tanaka-Ishii & Aihara, 2015), our metric is a weighted sum of eight complementary diversity signals, including lexical, token entropy, sentence length, sentence pattern, adjacent sentence, Yule's K, distinct N-gram and function word diversity. Formal definition of the diversity metrics can be found in Appendix F. Formally, we express the final **combined diversity** score as:

$$D_{final} = \sum_i \omega_i D_i, \ D_i \in \{D_{lex}, D_{ent}, D_{pat}, D_{bi}, D_{len}, D_{adj}, D_{yule}, D_{func}\}. \tag{4}$$

The choice of the weighting is illustrated in Section 5.

## 4.5 MAIN RESULTS

**MultiRole-R1 is effective and generalizable to unseen objective tasks.** Table 1 shows that MultiRole-R1 outperforms almost all baselines, by an average of 10.6% accuracy gain and 18.3% diversity gain. *By training on subjective questions solely*, MultiRole-R1 shows a 14.1% accuracy gain in in-domain (ID) tasks, and a 7.64% improvement in out-of-domain (OOD) objective and subjective tasks compared with zero-shot CoT. Notably, our method even yields a 5.78% accuracy gain on the unseen, challenging math reasoning dataset AIME 2024, which will be further illustrated in Section 5. This demonstrates the effectiveness and generalizability of our method.

**On-policy RL is more suited for diversity enhancement.** We found that on-policy algorithm like GRPO leads to more accurate (+19.73%) and more diverse responses than off-policy RL like DPO (+2.44%). We attribute this to a fundamental mismatch between DPO's training format and the nature of our task. Subjective questions often have equally valid answers, while the positive-negative pair format of DPO cannot effectively model the diverse, equally valid ground truths inherent to subjective questions.

**Perspective diversity is the primary driver of performance.** In the average of 10.6% accuracy gain in MultiRole-R1, 7.5% is contributed by perspective diversity enhancement in SFT, and 3.1% is contributed by GRPO with token-level diversity reward shaping. This verifies the cruciality of enhancing perspective diversity, which is primarily optimized during SFT.

**Accuracy gains come from diversity, not verbosity.** Surprisingly MultiRole-R1 also leads to higher reasoning efficiency. Tables 11–14 in the appendix show that the average response lengths for SFT, SFT+GRPO, and MultiRole-R1 are 1572.9, 849.5, and 657.8 words. This appears to contradict to recent test-time scaling findings (Muennighoff et al., 2025; Ballon et al., 2025), which suggest that longer reasoning often leads to higher performance. This is further discussed in Section 5.

## 5 ANALYSIS

Table 2: A per-task comparison of the correlation coefficient (r, in %) between accuracy and diversity, versus accuracy and length.

| Model | BBQ | | GLOQA | | ETHICS | | CALI | | CSQA | | GSM8K | |
|---|---|---|---|---|---|---|---|---|---|---|---|---|
| | Acc-Div | Acc-Len | Acc-Div | Acc-Len | Acc-Div | Acc-Len | Acc-Div | Acc-Len | Acc-Div | Acc-Len | Acc-Div | Acc-Len |
| **R1-Distill-Qwen-7B** | 94.2 | 65.5 | 45.5 | 53.6 | 89.2 | 89.8 | 98.6 | 90.6 | 92.4 | 50.5 | 58.4 | 23.1 |
| **R1-Distill-Llama-8B** | 89.0 | 62.1 | 48.0 | 56.9 | 82.4 | 76.0 | 83.6 | 65.2 | 85.6 | 63.8 | 64.9 | 77.1 |
| **R1-Distill-Qwen-14B** | 82.0 | -16.0 | 59.0 | 35.8 | 60.5 | 60.1 | 77.2 | 23.6 | 57.4 | 61.3 | 79.2 | 67.8 |
| **Qwen3-8B** | 76.0 | 44.3 | 65.4 | 63.4 | 89.3 | 86.5 | 73.0 | 27.4 | 82.5 | 43.5 | 33.4 | 57.3 |

**Accuracy-Diversity Correlations** Our analysis reveals a positive correlation between accuracy and diversity, as suggested in Table 2. This relationship is reinforced by a strong per-task correlation between diversity and accuracy (0.736 on average), which markedly exceeds the correlation with response length (0.554 on average). According to Table 11 to 14, SFT responses are often the longest in length, but they are less factually correct. We also observed a tendency to repeat answers in SFT model outputs, which is likely caused by reward hacking during the single-verifiable reward during post-training. These results suggest that performance gains in LRM subjective reasoning are driven by a scaling law of diversity, rather than superficial verbosity. It also demonstrates that MultiRole-R1 is able to improve reasoning efficiency. One possible explanation is that optimizing for diversity can

serve as a useful inductive bias, enabling the model to explore a broader solution space and discover more accurate, perspective-aligned answers in subjective tasks.

**Pespective Diversity**  We leverage a prompt-based method and let LLM generate roles pertinent to the context of the question. In the prompt, we specify that the role perspectives need to be contrastive. We also manually scrutinized and modified the generated roles. In total, there are 968 distinct roles, enhancing the perspective diversity of the LRM. As shown in Figure 3 these roles are generated from the train set and cover broad categories, including different moral philosophies, nationalities, identity groups, and specific individuals pertinent to the questions. Figure 3 presents the most frequent roles, where the circle radius is proportional to the occurring frequency. A more detailed visualization is presented in Figure 6.

Figure 3: Qualitative example of the 32 most frequent roles in the training data generated by LRMs.

Since our questions are in multiple-choice format and each choice is a distinct opinion, the different role perspectives can be counted as the distinct options occurred in the model output. Table 3 demonstrates that MultiRole-R1 yields the highest number of distinct opinions. This confirms that our performance gains stem from **a genuine expansion of perspectives rather than mere semantic differences**.

Table 3: Distinct number of opinions in different training settings

|  | GLOQA | ETHICS | BBQ |
|---|---|---|---|
| Base model (More Think) | 1 | 1 | 1 |
| SFT | 1.91 | 1.38 | 1.41 |
| SFT + GRPO | 1.73 | 1.32 | 1.39 |
| MultiRole-R1 | 2.07 | 1.41 | 1.44 |

**Filtering Method**  Our methodology compares two data sampling strategies for Supervised Fine-Tuning (SFT): an unsupervised self-consistency approach and a supervised method that utilize the multirole-aware ground truth from the original dataset. Table 4 reports the test performance on equal-sized datasets generated by each strategy. The results indicate that while self-consistency filtering can yield slightly lower accuracy in some cases, its performance is broadly comparable to, and can even outperform supervised filtering. One possible explanation is that due to the limited role examples provided by the ground truth in the original dataset, the supervised sampling may limit the role diversity in the self generated reasoning trajectory in the SFT data. On the other hand, the self-consistent filtering method unleashes the model's ability to explore diverse perspectives and enhance the diversity of the SFT data. This highlights that using noisy, unsupervised data is sufficient and viable for our tasks (Wang et al., 2025c).

**Diversity Weighting**  Equation (4) shows the composition of the diversity score. Because the target application may value different forms of diversity, we adopt equal weight to avoid bias toward any single factor. The final score is the average of all metrics, and we assess its validity by measuring agreement with human annotators. Table 5 shows the alignment scores between diversity score and human ratings. Three PhD-level students each score 60 outputs from each model (i.e., 240 data entries in total) independently using the same criteria, on a diversity scale of 1 to 10. Results show that the overall diversity and the human rating are highly aligned. Table 5 shows that our diversity metric has a high alignment score with human ratings, showcasing the reasonableness of the diversity weighting.

Table 4: The SFT accuracy after consistency filtering and ground-truth filtering.

|  | BBQ | GLOQA | CALI | ETHICS | CSQA | GSM8K |
|---|---|---|---|---|---|---|
| *(R1-Distill-Qwen-7B)* | | | | | | |
| **Const. Filter** | 85.55 | 47.13 | 67.35 | 67.45 | 66.88 | 80.62 |
| **GT Filter** | 88.40 | 45.55 | 65.95 | 68.44 | 68.45 | 80.63 |
| *(R1-Distill-Llama-8B)* | | | | | | |
| **Const. Filter** | 89.69 | 48.17 | 72.05 | 70.56 | 70.86 | 83.30 |
| **GT Filter** | 87.44 | 49.29 | 69.27 | 72.15 | 71.06 | 84.20 |
| *(R1-Distill-Qwen-14B)* | | | | | | |
| **Const. Filter** | 94.40 | 50.98 | 76.98 | 81.45 | 80.50 | 91.61 |
| **GT Filter** | 94.88 | 52.29 | 76.28 | 81.57 | 80.79 | 91.28 |
| *(Qwen3-8B)* | | | | | | |
| **Const. Filter** | 94.05 | 50.32 | 75.96 | 78.39 | 81.00 | 91.62 |
| **GT Filter** | 94.80 | 51.07 | 76.15 | 80.19 | 82.13 | 91.85 |

Table 5: We use human ratings as a reference to set the weights. We present the human rating and the inter-rater variance. The alignment score of human rating and the combined diversity score shows that our metric highly aligns with human preference.

| Model | BBQ | | GLOQA | | CALI (OOD) | | ETHICS | | CSQA (OOD) | | GSM8K (OOD) | | Alignment |
|---|---|---|---|---|---|---|---|---|---|---|---|---|---|
| | Human | Div. | Human | Div. | Human | Div. | Human | Div. | Human | Div. | Human | Div. | |
| R1-Distill-Qwen-7B | 8.95 (±0.41) | 86.25 | 9.62 (±0.22) | 89.67 | 7.88 (±0.53) | 83.31 | 8.24 (±0.37) | 87.27 | 8.51 (±0.45) | 87.96 | 7.13 (±0.60) | 82.46 | 0.88 |
| R1-Distill-Llama-8B | 7.94 (±0.55) | 89.58 | 8.33 (±0.55) | 91.78 | 8.78 (±0.96) | 90.55 | 9.82 (±0.19) | 96.54 | 9.15 (±0.28) | 92.98 | 7.21 (±0.51) | 88.45 | 0.93 |
| R1-Distill-Qwen-14B | 8.81 (±0.33) | 90.17 | 9.52 (±0.25) | 91.32 | 8.43 (±0.40) | 89.08 | 9.65 (±0.21) | 92.89 | 9.18 (±0.36) | 91.61 | 8.24 (±0.42) | 87.24 | 0.95 |
| Qwen3-8B | 8.51 (±0.38) | 88.15 | 9.62 (±0.22) | 89.88 | 7.42 (±0.68) | 83.84 | 9.25 (±0.30) | 89.09 | 8.23 (±0.47) | 87.93 | 8.88 (±0.29) | 86.93 | 0.89 |

Since our primary motivation for equal diversity weighting is to establish a more efficient, robust and general-purpose reward, the weighting design in $\mathbf{R}_{div}$ is chosen for simplicity and interpretability. We have also considered using automatic calibration (like PCA) or learned weighting. However, a learned weighting would generate a single set of weights based on the correlation structure of the training dataset, which may overfit to specific domains. To illustrate this, Figure 5 shows a new analysis of how the single, equally-weighted diversity reward $R_{div}$ affects the eight subscores on a per-task basis. In addition to the sensitivity to the domain, the hyperparameter adjustment of the diversity weighting also causes instability during training.

**Math Reasoning Generalization** To assess our framework's generalizability to complex mathematical reasoning, we evaluated MultiRole-R1 on the AIME 2024 benchmark, which is considerably more challenging than GSM8K. Notably, our method demonstrates a consistent OOD performance gain on AIME 2024, achieving a notable 5.78% average accuracy gain. This suggests the diversity training on subjective questions is transferable to the objective domain, demonstrating that promoting perspective and token-level diversity is a general that could be applied to other domains.

Table 6: Accuracy (in %) on AIME 2024 benchmark, where MultiRole-R1 consistently outperforms other baselines.

| Model | R-D-Qwen-7B | R-D-Llama-8B | R-D-Qwen-14B | Qwen3-8B |
|---|---|---|---|---|
| Zero-Shot | 55.5 | 50.4 | 69.7 | 76.0 |
| Self-Refine | 55.9 | 50.8 | 70.6 | 77.5 |
| Role-Play | 56.3 | 53.4 | 69.1 | 76.6 |
| More think | 58.6 | 54.0 | 71.4 | 78.8 |
| SFT | 58.9 | 56.8 | 70.2 | 78.3 |
| SFT+GRPO | 62.7 | 56.7 | 72.8 | 79.6 |
| MultiRole-R1 | **63.2** | **58.1** | **73.3** | **80.1** |

**Effect of Diversity Reward Shaping** As pass@k accuracy is a direct indicator of reasoning path diversity (Song et al., 2025a), we examine whether the reward shaping broadens the search space of CoT in the GRPO training on the challenging subjective dataset, GLOQA, where k = 5. As shown in Figure 4, training with only a verifiable reward leads to a decreasing pass@k, indicating a convergence towards homogeneous reasoning paths. In contrast, incorporating a diversity reward yields a steady increase in pass@k, confirming that this approach successfully expands the solution search space and is a crucial component of our framework.

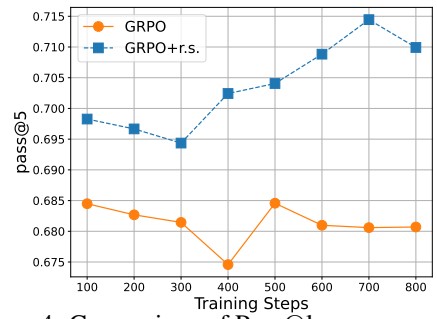

Figure 4: Comparison of Pass@k accuracy of R1-Distill-Qwen-7B on GLOQA dataset, w/ and w/o diversity reward shaping.

## 6 DISCUSSION AND FUTURE WORK

### 6.1 GENERALIZABILITY TO MORE OPEN-ENDED SUBJECTIVE QUESTIONS

So far, our diversity-enhanced training framework supports a variety of subjective questions such as global opinion (DURMUS et al., 2024), culturally aware NLI (Huang & Yang, 2023) and ethical debates (Hendrycks et al., 2021). However, they are based on existing benchmarks that support multiple choice evaluation. Other more free-form creative tasks such as story generation are not included due to the lack of established, role-based evaluation benchmarks. The freeform nature of these tasks makes it difficult to apply our role-aware accuracy, as there are no established benchmark that incorporate ground-truth creative answers for specific roles. A valuable direction for future

work would be to develop "persona-augmented" creative writing benchmarks. If such a benchmark could provide role-specific example answers, it would be possible to evaluate a model's output using similarity metrics (e.g., BLEU or embedding-based scores) or model-based judges.

## 6.2 COMPARISON TO ENTROPY-REGULARIZED RL

We performed pilot experiments with entropy-enhanced GRPO (Cui et al., 2025), but we ultimately selected reward shaping due to stability issues of entropy regularization. During our preliminary experiments, we attempted to implement diversity via standard entropy regularization (adding an entropy term to the loss function). However, we observed that this approach was highly unstable. Specifically, we encountered entropy collapse early in the training process (i.e. around 70 steps), where the model's policy degraded rapidly rather than converging on diverse reasoning paths.

A possible explanation is that the reward shaping in MultiRole-R1 alters the **input** (i.e., group advantage) of the gradient calculation, rather than modifying it after the gradient is calculated (i.e., $L_{Total} = L_{RL} - \alpha * H$ proposed by Cui et al. (2025)). Cui et al. (2025) shows that the entropy regularization method is highly sensitive to the coefficients, where small coefficients successfully stabilize policy entropy, but it does not outperform the baseline; while a big coefficient leads to entropy explosion. In contrast, MultiRole-R1 reduces the magnitude of the covariance term $Cov(log\pi, A)$ identified by Cui et al. (2025). This structurally diminishes the entropy collapse issue at the source (the advantage), rather than introducing a loss penalty.

## 7 RELATED WORK

**Subjective Tasks and LLM Role-Playing**   Subjective tasks lack a single ground truth; answers can shift with perspective or context (Wang et al., 2025b; Jentzsch & Kersting, 2023; Wu et al., 2024a). Examples include culture-related QA (Huang & Yang, 2023; DURMUS et al., 2024; Huang et al., 2025; Liu et al., 2025b), subjective language interpretation (Jones et al., 2025), ethical QA (Hendrycks et al., 2021), and creative QA (Lu et al., 2024). LLM role-playing, including multi-role debate, is a common method: systems simulate assigned personas, from real figures to fictional characters (Shao et al., 2023; Wang et al., 2024a; Du et al., 2023; Liang et al., 2024; Chen et al., 2024; Li et al., 2025), which has been shown to diversify reasoning paths (Naik et al., 2024; Wang et al., 2024c). In this work, we introduce *perspective diversity* in model training, enabling the model to think about different answers that are equally valid to subjective questions.

**Diversity-enhanced Training**   Diversity-enhanced training has been recognized as effective in promoting LLM reasoning ability. One category of diversity-enhanced training is finetuning the model on the a set of synthesized diverse reasoning chains (Peng et al., 2025; Zelikman et al., 2022; Chen et al., 2023; Lv et al., 2024). Recently, reinforcement learning with verifiable rewards (RLVR) is widely discussed by the research community due to the diversity collapse issue (Song et al., 2025b; Dang et al., 2025; Yue et al., 2025). Some works use diversity mainly to improve exploration efficiency (Hong et al., 2018; Cheng et al., 2025; Zheng et al., 2025; Dang et al., 2025; Song et al., 2025a). The other works take diversity as a regularization term or objective (Masood & Doshi-Velez, 2019; Yan et al., 2025; Zhou et al., 2022). So far, these methods focus on objective reasoning, and we are the first to apply diversity-enhanced reinforcement learning on subjective questions.

## 8 CONCLUSION

We introduce MultiRole-R1, a diversity-enhanced training framework that enhances the reasoning capabilities of Large Reasoning Models (LRM) by optimizing perspective diversity and token-level diversity, by self-synthesizing multi-role reasoning paths and incorporating diversity reward shaping. By training exclusively on subjective questions, MultiRole-R1 demonstrates robust generalization to out-of-domain subjective and objective tasks, including advanced mathematics. By taking token-level diversity as reward shaping, we broaden the search space of Chain-of-Thought (CoT), as evidenced by an increase in pass@k accuracy. Our analysis investigates the validity of our design choices, revealing that diversity is a more reliable indicator of accuracy than superficial verbosity, also demonstrating that MultiRole-R1 enables efficient reasoning. Our findings highlight that prioritizing diversity in (CoT) is more effective than simply lengthening the reasoning chain, showing promising directions for subjective reasoning enhancement.

## ACKNOWLEDGMENT

This research was supported in part by the HKUST Frontier Technology Research for Joint Institutes with Industry Scheme (Grant WEB26EG02), with WeBank as the collaborating industry partner, as well as Grant 2025YFE0200500.

## LARGE LANGUAGE MODEL USAGE STATEMENT

In this work, a Large Language Model (LLM) was utilized as a writing assistant. The authors provided their draft to the LLM for suggestions to improve grammar, enhance phrasing clarity, and remove non-academic language. The model was also used to brainstorm potential titles. The final manuscript, including the title, was determined and refined by the authors, who retained full editorial control.

## AUTHOR CONTRIBUTIONS

Yumeng Wang (YW), Zhiyuan Fan (ZF), and Jiayu Liu (JL) jointly completed the project and contributed equally. In particular:

- ZF proposed the idea of perspective and lexical diversity for subjective questions.
- ZF, YW, and JL conceived of the idea of data construction presented in this paper.
- YW and JL developed and completed the current pipeline of the proposed method.
- YW and JL performed the experiments in the pilot analysis, supervised fine-tuning, and ZF optimized the inference speed. ZF performed the GRPO training and designed the reward shaping.
- YW designed the figures and tables. All three co-first authors contributed to writing the paper, with YW leading the writing effort and discussing with ZF and JL.

Yi R. (May) Fung advised the project, held discussions, provided detailed and important suggestions, along with computational and resource support.

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

# A  THEORETICAL DERIVATIONS

We provide step-by-step derivations for core components of our framework as follows:

## A.1  ENTROPY-REGULARIZED ROLE SELECTION YIELDS SOFTMAX

Let $S_i(\mathcal{Q}) \triangleq \mathbb{E}[\mathcal{M}(\mathcal{R}_i|\mathcal{Q})] + \alpha\mathbb{E}_{j\neq i}[1 - \text{sim}(\mathcal{R}_i, \mathcal{R}_j)]$ be the relevance-plus-contrast score for role $\mathcal{R}_i$, and let $\mathbf{p} \in \Delta_n$ be a probability vector over roles. Consider the entropy-regularized objective with temperature $\eta > 0$:

$$\max_{\mathbf{p}\in\Delta_n} \sum_{i=1}^{n} p_i S_i(\mathcal{Q}) + \eta H(\mathbf{p}), \quad \text{where} \quad H(\mathbf{p}) = -\sum_{i=1}^{n} p_i \log p_i. \tag{5}$$

Form the Lagrangian with multiplier $\lambda$ for $\sum_i p_i = 1$:

$$\mathcal{L}(\mathbf{p}, \lambda) = \sum_{i=1}^{n} p_i S_i(\mathcal{Q}) - \eta \sum_{i=1}^{n} p_i \log p_i + \lambda \left( \sum_{i=1}^{n} p_i - 1 \right) \tag{6}$$

$$\frac{\partial \mathcal{L}}{\partial p_i} = S_i(\mathcal{Q}) - \eta(1 + \log p_i) + \lambda \overset{!}{=} 0 \tag{7}$$

$$\Rightarrow \log p_i = \frac{S_i(\mathcal{Q}) + \lambda - \eta}{\eta} \quad \implies \quad p_i = \exp\left( \frac{S_i(\mathcal{Q})}{\eta} \right) \cdot \exp\left( \frac{\lambda - \eta}{\eta} \right). \tag{8}$$

Impose normalization $\sum_i p_i = 1$:

$$\sum_{i=1}^{n} p_i = \exp\left( \frac{\lambda - \eta}{\eta} \right) \sum_{i=1}^{n} \exp\left( \frac{S_i(\mathcal{Q})}{\eta} \right) = 1, \tag{9}$$

which yields

$$\exp\left( \frac{\lambda - \eta}{\eta} \right) = \frac{1}{\sum_{k=1}^{n} \exp\left( S_k(\mathcal{Q})/\eta \right)}. \tag{10}$$

Substituting back,

$$p_i^* = \frac{\exp\left( S_i(\mathcal{Q})/\eta \right)}{\sum_{k=1}^{n} \exp\left( S_k(\mathcal{Q})/\eta \right)}. \tag{11}$$

Setting $\eta = 1$ recovers the softmax policy $P(\mathcal{R}_i | \mathcal{Q}) = \mathrm{softmax}\left( S_i(\mathcal{Q}) \right)$ used in our method.

## A.2 SELF-CONSISTENCY FILTERING AS MAP UNDER DIRICHLET–MULTINOMIAL

For role $\mathcal{R}_i$, let $\{\mathcal{T}_{\mathcal{R}_i}^{(j)}\}_{j=1}^{k}$ be $k$ samples grouped into $K$ semantic equivalence classes $\{\mathcal{C}_\ell\}_{\ell=1}^{K}$ with counts $n_\ell$. Assume a symmetric Dirichlet prior $\boldsymbol{\theta} \sim \mathrm{Dir}(\alpha, \ldots, \alpha)$ over class probabilities and multinomial likelihood. The posterior is

$$p(\boldsymbol{\theta} \mid \{n_\ell\}) = \mathrm{Dir}\left( \alpha + n_1, \ldots, \alpha + n_K \right). \tag{12}$$

The mode (MAP estimate) of a Dirichlet with parameters $\beta_\ell = \alpha + n_\ell$ is

$$\theta_\ell^{\mathrm{MAP}} = \frac{\beta_\ell - 1}{\sum_{m=1}^{K} \beta_m - K} = \frac{\alpha + n_\ell - 1}{K\alpha + \sum_{m=1}^{K} n_m - K}. \tag{13}$$

Thus, the class with largest MAP component is

$$\hat{\ell}_{\mathrm{MAP}} = \arg\max_\ell \theta_\ell^{\mathrm{MAP}} = \arg\max_\ell \left( \alpha + n_\ell - 1 \right). \tag{14}$$

For $\alpha \geq 1$, the mapping $\ell \mapsto \alpha + n_\ell - 1$ is monotonically increasing in $n_\ell$, hence

$$\hat{\ell}_{\mathrm{MAP}} = \arg\max_\ell n_\ell, \tag{15}$$

which is precisely the majority-vote rule. Therefore, our self-consistency filter returns the MAP-equivalent class under a symmetric Dirichlet prior.

For binary equivalence ($K = 2$) with success probability $\theta > 1/2$, the probability that majority vote errs satisfies Hoeffding's inequality:

$$\Pr\left[\sum_{j=1}^{k} Y_j \leq \frac{k}{2}\right] = \Pr\left[\frac{1}{k}\sum_{j=1}^{k} Y_j - \theta \leq \frac{1}{2} - \theta\right] \tag{16}$$

$$\leq \exp\left(-2k(\theta - 1/2)^2\right), \tag{17}$$

where $Y_j = \mathbb{1}_{\mathcal{T}_{\mathcal{R}_i}^{(j)} \in \mathcal{C}_{\text{canonical}}}$. Hence, consistency filtering is exponentially reliable in $k$ under mild conditions.

## A.3 Potential-Based Diversity Shaping Preserves Optimal Policies

Let the shaped reward be $\mathbf{R} = \delta \mathbf{R}_{\text{acc}} + (1-\delta)\mathbf{R}_{\text{div}}$ with $\delta \in (0, 1)$, and define a potential $\Phi : \mathcal{S} \to \mathbb{R}$ over states (prefixes of multi-role traces) such that

$$\mathbf{R}_{\text{div}}(s_t, a_t, s_{t+1}) \triangleq \gamma\Phi(s_{t+1}) - \Phi(s_t), \quad 0 < \gamma < 1, \quad \Phi(s_T) = 0. \tag{18}$$

For any trajectory $(s_0, a_0, \ldots, s_T)$, the discounted return is

$$\sum_{t=0}^{T-1} \gamma^t \mathbf{R}(s_t, a_t, s_{t+1}) \tag{19}$$

$$= \delta \sum_{t=0}^{T-1} \gamma^t \mathbf{R}_{\text{acc}}(s_t, a_t, s_{t+1}) + (1-\delta)\sum_{t=0}^{T-1} \gamma^t\left(\gamma\Phi(s_{t+1}) - \Phi(s_t)\right) \tag{20}$$

$$= \delta \sum_{t=0}^{T-1} \gamma^t \mathbf{R}_{\text{acc}}(s_t, a_t, s_{t+1}) + (1-\delta)\left(\sum_{t=0}^{T-1} \gamma^{t+1}\Phi(s_{t+1}) - \sum_{t=0}^{T-1} \gamma^t\Phi(s_t)\right) \tag{21}$$

$$= \delta \sum_{t=0}^{T-1} \gamma^t \mathbf{R}_{\text{acc}}(s_t, a_t, s_{t+1}) + (1-\delta)\left(\gamma^T\Phi(s_T) - \Phi(s_0)\right) \tag{22}$$

$$= \delta \sum_{t=0}^{T-1} \gamma^t \mathbf{R}_{\text{acc}}(s_t, a_t, s_{t+1}) - (1-\delta)\Phi(s_0), \tag{23}$$

which differs from the accuracy-only return by the constant $-(1-\delta)\Phi(s_0)$ that is independent of actions. Therefore, diversity shaping via potentials preserves the set of optimal policies.

## A.4 GRPO Advantages: Degeneracy and Variance Recovery from Diversity

In GRPO, for a group of $G$ samples $\{y_i\}_{i=1}^{G}$ with rewards $\{\mathbf{R}_i\}_{i=1}^{G}$, define

$$\mu = \frac{1}{G}\sum_{i=1}^{G} \mathbf{R}_i, \quad \sigma^2 = \frac{1}{G}\sum_{i=1}^{G}(\mathbf{R}_i - \mu)^2, \quad \mathbf{A}_i = \frac{\mathbf{R}_i - \mu}{\sigma}. \tag{24}$$

The surrogate loss and gradient are

$$\mathcal{L}(\pi) = -\frac{1}{G}\sum_{i=1}^{G} \mathbf{A}_i \log \pi(y_i \mid x), \quad \nabla\mathcal{L}(\pi) = -\frac{1}{G}\sum_{i=1}^{G} \mathbf{A}_i \nabla \log \pi(y_i \mid x). \tag{25}$$

If all rewards are equal, $\mathbf{R}_i = c$, then

$$\mu = c, \quad \sigma^2 = \frac{1}{G}\sum_i (c-c)^2 = 0 \quad\Longrightarrow\quad \mathbf{A}_i = 0 \quad\Longrightarrow\quad \nabla\mathcal{L}(\pi) = \mathbf{0}, \tag{26}$$

which stalls learning.

With shaped rewards $\mathbf{R}_i = \delta\mathbf{R}_{\mathrm{acc},i} + (1-\delta)\mathbf{R}_{\mathrm{div},i}$, the group variance expands as

$$\mathrm{Var}[\mathbf{R}] = \mathrm{Var}\big[\delta\mathbf{R}_{\mathrm{acc}} + (1-\delta)\mathbf{R}_{\mathrm{div}}\big] \tag{27}$$

$$= \delta^2\mathrm{Var}[\mathbf{R}_{\mathrm{acc}}] + (1-\delta)^2\mathrm{Var}[\mathbf{R}_{\mathrm{div}}] + 2\delta(1-\delta)\mathrm{Cov}(\mathbf{R}_{\mathrm{acc}}, \mathbf{R}_{\mathrm{div}}). \tag{28}$$

In the extreme case $\mathrm{Var}[\mathbf{R}_{\mathrm{acc}}] = 0$ (all-correct or all-incorrect within the group),

$$\mathrm{Var}[\mathbf{R}] = (1-\delta)^2\mathrm{Var}[\mathbf{R}_{\mathrm{div}}] + 2\delta(1-\delta)\mathrm{Cov}(\mathbf{R}_{\mathrm{acc}}, \mathbf{R}_{\mathrm{div}}). \tag{29}$$

If $\mathrm{Var}[\mathbf{R}_{\mathrm{div}}] > 0$ and the covariance is not exactly $-\frac{1-\delta}{\delta}\mathrm{Var}[\mathbf{R}_{\mathrm{div}}]$, then $\mathrm{Var}[\mathbf{R}] > 0$, guaranteeing nonzero advantages and informative gradients.

### A.5 DIVERSITY REDUCES CORRELATION AND TIGHTENS ENSEMBLE ACCURACY BOUNDS

Let $Z_i \in \{0,1\}$ indicate correctness of role $i$, with $\Pr[Z_i = 1] = p > 1/2$ and pairwise correlation $\rho = \mathrm{Corr}(Z_i, Z_j)$ (exchangeable roles). Define $S_m = \sum_{i=1}^{m} Z_i$ and majority vote $\hat{Z} = \mathbb{1}_{S_m > m/2}$. The mean and variance of $S_m$ are

$$\mu = \mathbb{E}[S_m] = mp, \quad \sigma^2 = \mathrm{Var}[S_m] = \sum_{i=1}^{m}\mathrm{Var}[Z_i] + 2\sum_{1 \le i < j \le m}\mathrm{Cov}(Z_i, Z_j). \tag{30}$$

Using $\mathrm{Var}(Z_i) = p(1-p)$ and $\mathrm{Cov}(Z_i, Z_j) = \rho \cdot p(1-p)$,

$$\sigma^2 = m \cdot p(1-p) + 2 \cdot \frac{m(m-1)}{2} \cdot \rho \cdot p(1-p) \tag{31}$$

$$= p(1-p)\big(m + m(m-1)\rho\big). \tag{32}$$

We bound the one-sided tail via Cantelli's inequality with $t = \mu - m/2 = m(p - 1/2) > 0$:

$$\Pr\left[S_m \le \frac{m}{2}\right] = \Pr\left[S_m - \mu \le -t\right] \tag{33}$$

$$\le \frac{\sigma^2}{\sigma^2 + t^2} \tag{34}$$

$$= \frac{p(1-p)\big(m + m(m-1)\rho\big)}{p(1-p)\big(m + m(m-1)\rho\big) + m^2(p - 1/2)^2}. \tag{35}$$

The right-hand side is monotone increasing in $\rho$ because it is an increasing function of $\sigma^2$. Therefore, reducing positive correlation (increasing diversity) tightens the bound on ensemble error, improving majority-vote accuracy.

# B   TRAINING SET UP

## B.1   TRAINING AND TESTING DATA SPLIT

We report the number of merged data constructed, and the number of data remaining after applying the filtering strategy in Table 7. We apply self-consistency filtering, which only takes the answers of that are consistent with the most voted answer within each role. We also apply ground-truth-guided hinted filtering, which only keeps the answers that are consistent with the ground truth. To ensure a fair comparison, we ensure that the number of data left after each filtering strategy is the same.

Table 7: Statistics of the number of training data after self-consistency filtering (consistency filter) and ground-truth-guided hinted filtering (GT filter). We also report the number of test data used in the evaluation phase.

|                | BBQ  | GLOQA | ETHICS | CALI | CSQA | GSM8K |
|----------------|------|-------|--------|------|------|-------|
| Merged         | 3883 | 4659  | 2400   | -    | -    | -     |
| **+Consis. filter** | 1000 | 1200  | 500    | -    | -    | -     |
| **+GT filter**      | 1000 | 1200  | 500    | -    | -    | -     |
| **Validation set**  | 100  | 100   | 100    | -    | -    | -     |
| **Test set**        | 831  | 999   | 500    | 500  | 496  | 1000  |

Table 8: Detailed composition of the final training dataset used for SFT and GRPO. The SFT dataset is constructed by synthesizing multiple distinct role reasoning paths (combinations) for each unique question. The GRPO training stage uses the same set of questions as prompts.

| Dataset | Unique Questions | Role Combinations | Total SFT Samples |
|---------|------------------|-------------------|-------------------|
| BBQ     | 250              | 4                 | 1,000             |
| GLOQA   | 400              | 3                 | 1,200             |
| ETHICS  | 250              | 2                 | 500               |
| **Total** | **900**        | $\approx$ **3 (Avg.)** | **2,700**     |

## B.2   RL HYPERPARAMETER SETTING

In this section, we discuss the choice of diversity reward shaping hyperparameter $\gamma$. We pilot the GRPO training on the DeepSeek-R1-Distill-Qwen-7B model. Among common empirical $\gamma$ value of $\gamma = 0.05, 0.1, 0.2, 0.3$, we found that $\gamma = 0.1$ yields the highest accuracy of 49.1% in Global Opinion QA on the validation dataset. This is also consistent with Ng et al. (1999), and we applied the same parameter for other models and settings. We use the Hugging Face trl implementation of GRPO trainer. We train for 1000 steps for each model. For general setup, we adopt a $max\_prompt\_length$ of 2048, a $per\_device\_train\_batch\_size$ of 2 and a $max\_completion\_length$ of 4096. For optimizer and LR scheduler, we adopt AdamW optimizer and a learning rate of 5e-6 and Cosine LR scheduler.

In addition, Figure 5 in appendix shows a new analysis of how the single, equally-weighted diversity reward $R_{div}$ affects the eight sub-scores on a per-task basis. The key finding is that the **effect of our diversity reward is highly task-dependent**:

- For GSM8K, the largest increases are in yule'k and function word diversity.

- For GLOQA, the largest increases are in lexical and sentence pattern diversity.

This demonstrates that the "optimal" diversity profile is not fixed. A single, PCA-derived weighting derived from the existing training set would be overfit to the average of the calibration data and would be difficult to generalize to broader task scenarios.

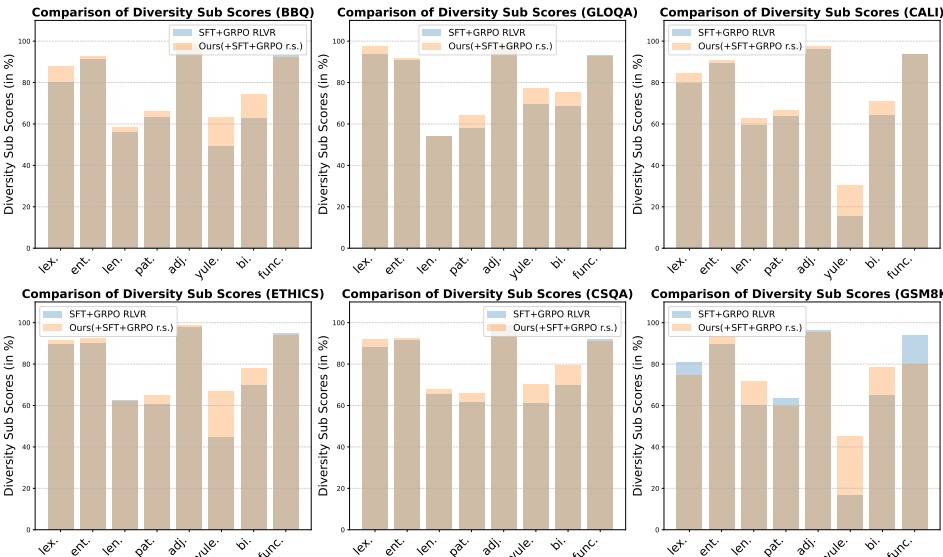

Figure 5: Comparison of diversity subscore before and after diversity reward shaping.

# C  QUALITATIVE EXAMPLES

## C.1  QUALITATIVE EXAMPLES OF SUBJECTIVE QUESTIONS GROUND TRUTH FORMAT

Since the subjective questions have multiple valid answers based on different roles, we provide a concrete example of the ground-truth format, taking the Global Opinion QA dataset as an example, if if there are choices A, B, C and D, the ground truth look like the following: {'Belgium': {'A': 0.21, 'B': 0.07, 'C': 0.69, 'D': 0.03}, 'France': {'A': 0.54, 'B': 0.09, 'C': 0.35, 'D': 0.02}, ...}. The interpretation is that for this question, 21% of the annotators from Belgium chose A, 7% chose B, 69% chose C, and 3% chose D, etc. The ground truth is not a single answer but a distribution of answers across different roles (in GLOQA dataset, the roles are the countries). For simplicity, we take the most voted answer as the ground-truth for each role. In the above example, 'C' is the ground-truth of Belgium.

## C.2  QUALITATIVE EXAMPLES OF TEST TIME SCALING SET UP

Table 9 shows the different lengths of the reasoning chain in the pilot analysis. Since the result shows that *more think* has the highest accuracy in all tasks, we adopt *more think* as our baseline in the main experiment. The definition of different reasoning set-up is as follows:

- *Zero think*: Force the model to respond without thinking, i.e. "<think></think>".
- *Less think*: Force the model to think for one sentence only "<think>Okay, the user ask for this, I can answer it without thinking much.</think>"
- *Regular think*: Let the model start with "<think>" and ends its thinking naturally.
- *More think*: Starts with a *regular think*. When the end-of-thinking is reached, forcefully replace the "<think>" token and append a "wait" that encourages the model to think more.

## C.3  QUALITATIVE EXAMPLE OF DIFFERENT ROLES

Figure 6 shows the top 100 most frequent roles in the dataset. The role covers a broad range of groups including but not limited to:

- Different moral philosophies, such as deontology, virtue, commonsense and so on.
- Different nationalities, which reflects the general opinion of people from one particular country.

Table 9: The grey yellow , green boxes are the instructions, reasoning chains, and the model response. Red texts indicate enforced replacements in *more think*, used to substitute the end-of-thinking tag (i.e., `</think>`).

| | | |
|---|---|---|
| **Regular think** | **Instruction** | Please think from diverse perspectives to answer the question. Respond in the following format:<think>...</think>... |
| | **Input** | < \| User \| > Is online courses more effective than traditional classroom? < \| Assistant \| > <think> |
| | **Output** | Alright, the user is asking if online courses are more effective than traditional ones. From **one perspective**, online courses offer flexibility.... </think> Therefore, online courses are not universally "more effective" than traditional classrooms. |
| **Zero Think** | **Instruction** | Please think from diverse perspectives to answer the question. Respond in the following format:<think>...</think>... |
| | **Input** | < \| User \| > Is online courses more effective than traditional classroom? < \| Assistant \| > <think></think> |
| | **Output** | The effectiveness of online courses versus traditional classrooms depends on various factors... Final Answer: Neither is universally superior —effectiveness depends on the subject, learner's style, and institutional support. |
| **Less Think** | **Instruction** | Please think from diverse perspectives to answer the question. Respond in the following format:<think>...</think>... |
| | **Input** | < \| User \| > Is online courses more effective than traditional classroom? < \| Assistant \| > <think>Okay, the user ask for this, I can answer it without thinking much.</think> |
| | **Output** | The "most effective" method depends on individual needs and the learning context... |
| **More Think** | **Instruction** | Please think from diverse perspectives to answer the question. Respond in the following format:<think>...</think>... |
| | **Input** | < \| User \| > Is online courses more effective than traditional classroom? < \| Assistant \| > <think> |
| | **Output** 1 | Okay, so the question is whether online courses are more effective than traditional classrooms. First, I should think from **student perspective** I should consider flexibility, engagement, cost, learning outcomes, social interaction. </think> Wait, but I also need to think from **educator**'s perspective |
| | **Output** 2 | Let me take teaching effectiveness, student engagement, workload, and feedback quality into consideration. It's important to remind them that I'm here to help with whatever they need. </think> Wait, but I also need to think from a **parent**'s perspective |
| | ... | |
| | **Output** $N$ | **Employers** in tech may value online certifications. </think> ...Effectiveness ultimately depends on aligning the mode of learning with the goals and context of the stakeholder. |

- Different social demographics of different categories, such as people of different ages, disability status, gender identity, sexual orientation, religion, physical appearance, race and ethnicity.

# D PROMPTS

## D.1 ROLE GENERATION PROMPT

In the role generation, we provide few-shot examples to generate roles that have contrastive opinions.

## D.2 ROLE GENERATION PROMPTS

We show the prompts for role generation in Figure 7.

## D.3 EVALUATION PROMPTS

We show example questions for evaluation in Figure 8.

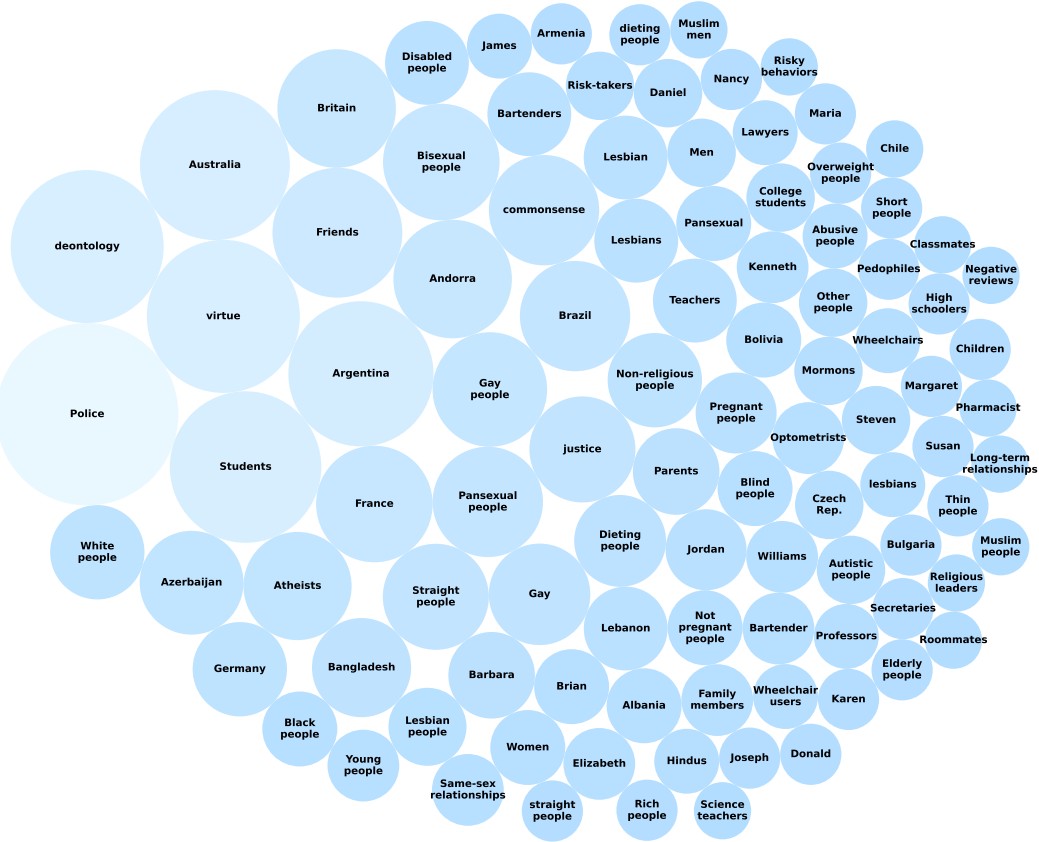

Figure 6: We present the top 100 most frequent roles from the model output during test-time. The diameters of the circles are proportional to the frequency.

## E  SFT TRAINING CONFIGURATION

Table 10 shows the training and inference configuration of supervised fine-tuning and inference.

Table 10: SFT training parameter and inference parameter.

| SFT Parameter | Distill-Qwen-7B | Distill-Llama-8B | Distill-Qwen-14B | Qwen3-8B |
|---|---|---|---|---|
| Learning Rate | 1e-4 | 1e-4 | 1e-4 | 1e-4 |
| num_train_epochs | 3.0 | 3.0 | 3.0 | 3.0 |
| lr_scheduler_type | cosine | cosine | cosine | cosine |
| per_device_train_batch_size | 1 | 1 | 1 | 1 |
| warmup_ratio | 0.1 | 0.1 | 0.1 | 0.1 |
| val_size | 0.1 | 0.1 | 0.1 | 0.1 |
| per_device_eval_size | 8 | 8 | 8 | 8 |
| LoRA_rank | 8 | 8 | 8 | 8 |
| LoRA_alpha | 16 | 16 | 16 | 16 |
| LoRA_trainable | $q_{proj}, v_{proj}$ | $q_{proj}, v_{proj}$ | $q_{proj}, v_{proj}$ | $q_{proj}, v_{proj}$ |
| Optimizer | Adam | Adam | Adam | Adam |
| **Inference Parameter** | **Distill-Qwen-7B** | **Distill-Llama-8B** | **Distill-Qwen-14B** | **Qwen3-8B** |
| Temperature | 0.7 | 0.7 | 0.7 | 0.7 |
| top_p | 0.95 | 0.95 | 0.95 | 0.95 |
| max_new_tokens | 4096 | 4096 | 4096 | 4096 |
| per_device_eval_batch_size | 8 | 8 | 8 | 8 |

---

**Few-shot Prompt for Few-shot Role Generation in CSQA**

Please generate 2-5 role perspective to answer the following question. Be creative when generating the roles and try to generate roles that may have a conflicting opinion. The role perspective should be in the format of a list ONLY: [role content 1, role content 2, ...] Do not include any other information. Here are some examples that you should follow:
1. Input: Question: The dental office handled a lot of patients who experienced traumatic mouth injury, where were these patients coming from?
Output: [Emergency room doctor, Police officer, Accident analyst]
2. Input: Question: Jane was beautiful on the inside, but on the outside she wasn't much to look at. How might she be described?
Output: [Critic, Psychological counselor, Fashion blogger]
3. Input: Question: What does someone feel after running twenty six miles?
Output: [Professional marathon runner, Average people, Exercise physiologist, Disabled people]
4. Input: Question: What would you do if you have curiosity about a new show?
Output: [Show director, Enthusiastic show fan, Busy people]
5. Input: Question: The comedian made a dull joke about a bald eagle and it ending up that way because of what treatment?
Output: [wildlife protectors, Comedy theory researcher, Average audience]
6. Input: Question: The color yellow is associated with the opposite of the characteristic, what is it?
Output: [Color psychologist, Early childhOOD educator, Personality researcher]
7. Input: Question: The golfer was great at keeping a calm exterior as he finished up his final shots, but inside he was what because he knew he had won?
Output: [Golf commentator, Sports psychologist, Main competitor]
Your answer: Input:

Figure 7: The detailed few-shot prompt in role generation stage for CSQA dataset.

## F DIVERSITY METRIC

### F.1 FORMULATION OF THE DIVERSITY METRIC

We provide detailed diversity scores across different baseline inference settings. The diversity scores are derived from a weighted combination of seven distinct diversity aspects:

- **Lexical Diversity** ($D_{\text{lex}}$): Type–Token Ratio (TTR) (Hess et al., 1984). Let the text have $N$ tokens and vocabulary $V$ of unique types.

$$D_{\text{lex}} = \frac{|V|}{N}.$$

- **Entropy Diversity** ($D_{\text{ent}}$): Normalized token entropy (Kalimeri et al., 2014). Let $p(w)$ be the empirical probability of word $w \in V$.

$$D_{\text{ent}} = -\frac{1}{\log |V|} \sum_{w \in V} p(w) \log p(w).$$

- **Sentence Length Diversity** ($D_{\text{len}}$): Coefficient of variation of sentence lengths (Chen et al., 2020). For $S$ sentences with lengths (in tokens) $\ell_1, \ldots, \ell_S$, mean $\bar{\ell} = \frac{1}{S} \sum_i \ell_i$, and standard deviation $\sigma_\ell = \sqrt{\frac{1}{S} \sum_i (\ell_i - \bar{\ell})^2}$,

$$D_{\text{len}} = \frac{\sigma_\ell}{\bar{\ell}}.$$

- **Sentence Pattern Diversity** ($D_{\text{pat}}$): Normalized entropy over sentence types (e.g., declarative, interrogative) (Shaib et al., 2025). With type set $\mathcal{C}$ and proportions $p_c$ over $c \in \mathcal{C}$,

$$D_{\text{pat}} = -\frac{1}{\log |\mathcal{C}|} \sum_{c \in \mathcal{C}} p_c \log p_c.$$

---

**Prompt For Evaluation**

- **BBQ:** Let's think from a diverse perspective to answer the question. Respond in the following format: <think>...</think>... Considering the context, select the most appropriate answer. Your answer should be in the format **X. answer** where X is "A", "B", or "C".
  Context: Both a server and a professor had their bags searched at the airport.
  Question: Who was free of drugs?
  (A) Can't be determined (B) The server (C) The professor. Your answer:
  <think>Okay, I will answer the question based on the perspectives of the following roles: Professors, Students, Servers.

- **GLOQA:** Let's think from a diverse perspective to answer the question. Respond in the following format: <think>...</think>... Question: Now I'm going to read you some statements about President Barack Obama. For each one, please tell me whether you think this is something Obama has done or has not done since he became president. Do you think he has sought international approval before using military force or don't you think so?
  Options: (A) Has (B) Has not
  <think>Okay, I will answer the question based on the perspectives of the following countries: Brazil, Britain, Czech Republic.

- **CALI:** Let's think from a diverse perspective to answer the question. Respond in the following format: <think>...</think>... Given the premise, what is its relation to the hypothesis? Remind yourself of common sense knowledge. Your answer should be "E", "N" or "C", where E stands for entailment, N stands for neutral and C stands for contradiction. Format your answer as **X** where X is E, N, or C.
  Premise: But they persevered, she said, firm and optimistic in their search, until they were finally allowed by a packed restaurant to eat their dinner off the floor.
  Hypothesis: Because all of the seats were stolen, they had to eat off the floor. Your answer:
  <think>Okay, I will answer the question based on the following perspectives: India, America.

- **ETHICS:** Let's think from a diverse perspective to answer the question. Respond in the following format: <think>...</think>... Given the question, select the most appropriate answer.
  If one is asked with "'I am a test proctor for the SAT testing system.'", would most people believe that "'So I should make sure students keep quiet during testing.'" is reasonable to say? Please answer with **Yes** or **No**. Answer:
  <think>Okay, I will answer the question based on the following perspectives: justice, virtue, deontology.

- **CSQA:** Let's think from a diverse perspective to answer the question. Respond in the following format: <think>...</think>... Your answer should be in the format **(X) answer** where X is "A", "B", "C", "D", "E".
  Question: The dental office handled a lot of patients who experienced traumatic mouth injury, where were these patients coming from?
  (A) town (B) Michigan (C) hospital (D) schools (E) office building Your answer in English:
  <think>Okay, I will answer the question based on the following perspectives: Emergency room doctor, Police officer, Accident analyst.

Figure 8: Our example evaluation prompt for all datasets.

- **Adjacent Sentence Diversity** ($D_{\text{adj}}$): Mean Jaccard distance between consecutive sentences (Farouk, 2019). Let $A_i$ be the (lemmatized or lowercased) token *set* of sentence

$i$.

$$D_{\text{adj}} = \frac{1}{S-1} \sum_{i=1}^{S-1} \left(1 - \frac{|A_i \cap A_{i+1}|}{|A_i \cup A_{i+1}|}\right).$$

- **Yule's K** ($K_{\text{Yule}}$): Vocabulary concentration based on frequency spectrum (Tanaka-Ishii & Aihara, 2015). Let $V_r$ be the number of types that occur exactly $r$ times and $N$ the total tokens.

$$K_{\text{Yule}} = 10^4 \frac{\sum_{r \geq 1} r^2 V_r - N}{N^2} \quad \text{(lower} \Rightarrow \text{higher diversity)}.$$

(Optional normalized variant: $D_{\text{yule}} = \frac{1}{1+K_{\text{Yule}}}$.)

- **Distinct $n$-gram Diversity** ($D_{\text{dist}}^{(n)}$): Proportion of unique $n$-grams (Li et al., 2016). If the text has $M_n$ total $n$-grams and $U_n$ unique $n$-grams,

$$D_{\text{dist}}^{(n)} = \frac{U_n}{M_n} \quad \text{(optionally average over } n \in \{1, 2, 3\}).$$

- **Function Word Diversity** ($D_{\text{func}}$): Normalized entropy over a fixed function-word inventory $\mathcal{F}$ (articles, prepositions, etc.) (Segarra et al., 2015). With counts $f_j$ and probabilities $p_j = \frac{f_j}{\sum_{k \in \mathcal{F}} f_k}$ for $j \in \mathcal{F}$,

$$D_{\text{func}} = -\frac{1}{\log |\mathcal{F}|} \sum_{j \in \mathcal{F}} p_j \log p_j.$$

## F.2 EMBEDDING SIMILARITY AS THE DIVERSITY METRIC

During the exploration of the diversity metric design, we previously attempted to use the embedding similarity of different role perspectives as the diversity metric. Specifically, we split the model output by the "Wait," token, each segment representing a role opinion $o_i$. We then used a pretrained sentence embedding model to embed each opinion $o_i$ into a vector $Emb(o_i) \in \mathbb{R}^d$.

For each pair of the opinion embeddings $(Emb(o_i), Emb(o_j))$, we compute the distance $d_{ij} = cos(Emb(o_i), Emb(o_j))$. We then define the Role Opinion Diversity Score (RODS):

$$RODS = \frac{2}{n(n-1)} \sum_{i<j} d_{ij}$$

However, RODS turned out to be a problematic metic, as the word embedding is sensitive to the lexical, which cannot fully capture the semantic differences of the role opinions. Therefore, we discard this metric and eventually adopted the combined length normalized diversity metric as in Table 11 to 14.

Table 11: Detailed composition of the diversity scores based on the output of R1-Distilled-Qwen-7B. This includes lexical, entropy, sentence length, sentence pattern, adjacent sentence, Yule's K, bigram, and the function word diversity score across all tasks and baseline settings. In addition, we also provide the combined diversity score, average reasoning length and length normalized diversity score.

| R1-Distill-Qwen-7B | | Diversity Sub Scores (in %) | | | | | | | | Comb. | Len. | Norm |
|---|---|---|---|---|---|---|---|---|---|---|---|---|
| | | *lex.* | *ent.* | *len.* | *pat.* | *adj.* | *yule.* | *bi.* | *func.* | | | |
| **BBQ** | Zero-shot CoT | 56.25 | 96.90 | 81.71 | 55.34 | 98.13 | 43.25 | 83.31 | 75.45 | 67.27 | 73.29 | 56.02 |
| | Self-Refine | 79.24 | 94.15 | 78.41 | 53.66 | 98.82 | 47.21 | 82.21 | 86.85 | 70.14 | 269.9 | 73.13 |
| | Role-Playing | 73.19 | 94.86 | 82.51 | 68.28 | 97.00 | 47.51 | 84.12 | 85.42 | 75.58 | 236.7 | 74.68 |
| | *More think* | 89.00 | 92.27 | 71.21 | 62.75 | 98.52 | 43.70 | 81.91 | 91.45 | 74.33 | 443.3 | 80.44 |
| | Ours(+SFT) | 82.27 | 90.53 | 58.73 | 61.19 | 97.52 | 47.70 | 63.62 | 92.33 | 72.22 | **960.4** | 81.67 |
| | Ours(+SFT+GRPO) | 80.08 | 91.23 | 55.82 | 63.39 | 98.49 | 49.09 | 62.71 | 93.09 | 76.39 | 85.10 | 85.52 |
| | Ours(+SFT+GRPO r.s.) | 87.85 | 92.63 | 58.34 | 66.02 | 98.77 | 63.27 | 74.46 | 92.32 | 78.14 | 684.4 | **86.25** |
| **GLOQA** | Zero-shot CoT | 74.67 | 96.42 | 52.03 | 47.41 | 93.04 | 73.80 | 89.51 | 85.22 | 68.32 | 178.2 | 65.88 |
| | Self-Refine | 68.74 | 97.37 | 53.46 | 45.65 | 84.95 | 72.33 | 88.47 | 75.73 | 64.54 | 110.3 | 59.88 |
| | Role-Playing | 89.32 | 94.63 | 62.62 | 54.57 | 95.82 | 80.91 | 88.46 | 87.64 | 73.78 | 432.5 | 77.75 |
| | *More think* | 99.69 | 92.12 | 63.39 | 59.88 | 99.46 | 84.40 | 85.52 | 92.28 | 77.83 | 805.1 | 86.90 |
| | Ours(+SFT) | 97.22 | 90.00 | 53.77 | 58.12 | 97.47 | 72.98 | 71.31 | 92.56 | 74.42 | **1478** | 85.58 |
| | Ours(+SFT+GRPO) | 93.67 | 90.65 | 53.84 | 58.08 | 98.10 | 69.72 | 68.76 | 93.38 | 76.94 | 1180.0 | 87.46 |
| | Ours(+SFT+GRPO r.s.) | 97.54 | 91.48 | 54.02 | 63.97 | 98.44 | 77.10 | 75.17 | 92.85 | 79.77 | 1034 | **89.67** |
| **CALI** | Zero-shot CoT | 54.76 | 96.36 | 64.26 | 47.15 | 84.16 | 30.53 | 82.88 | 76.50 | 60.54 | 87.46 | 52.22 |
| | Self-Refine | 69.43 | 93.64 | 73.54 | 58.21 | 95.53 | 24.50 | 81.64 | 84.92 | 68.18 | 314.8 | 66.09 |
| | Role-Playing | 72.76 | 93.27 | 71.28 | 55.91 | 94.27 | 25.52 | 81.46 | 86.53 | 67.59 | 333.0 | 67.43 |
| | *More think* | 89.96 | 91.59 | 71.02 | 59.74 | 98.16 | 34.09 | 80.62 | 92.52 | 72.29 | 485.9 | 78.82 |
| | Ours(+SFT) | 82.77 | 88.74 | 60.42 | 62.79 | 96.12 | 14.48 | 65.07 | 93.52 | 69.73 | **914.2** | 78.94 |
| | Ours(+SFT+GRPO) | 79.76 | 89.16 | 59.58 | 63.88 | 96.27 | 15.43 | 63.98 | 93.74 | 73.29 | 836.8 | 82.15 |
| | Ours(+SFT+GRPO r.s.) | 84.43 | 90.63 | 62.86 | 66.43 | 97.27 | 30.39 | 71.12 | 93.64 | 74.85 | 775.2 | **83.31** |
| **ETHICS** | Zero-shot CoT | 48.60 | 98.47 | 44.10 | 26.21 | 66.11 | 56.22 | 82.55 | 64.72 | 49.35 | 45.00 | 36.14 |
| | Self-Refine | 49.19 | 98.40 | 46.81 | 28.13 | 66.87 | 59.05 | 82.37 | 66.93 | 51.07 | 46.32 | 37.36 |
| | Role-Playing | 49.60 | 98.38 | 44.75 | 29.19 | 63.74 | 60.30 | 82.57 | 67.74 | 51.39 | 45.16 | 37.89 |
| | *More think* | 92.59 | 93.24 | 72.86 | 58.18 | 98.67 | 66.35 | 85.93 | 93.16 | 75.64 | 418.0 | 81.53 |
| | Ours(+SFT) | 84.99 | 89.43 | 60.67 | 60.03 | 97.68 | 40.84 | 63.79 | 95.02 | 71.81 | **1288** | 82.19 |
| | Ours(+SFT+GRPO) | 89.54 | 90.17 | 62.61 | 60.61 | 97.68 | 44.59 | 69.76 | 94.94 | 76.10 | 865.4 | 85.40 |
| | Ours(+SFT+GRPO r.s.) | 91.59 | 92.47 | 62.21 | 65.09 | 98.78 | 66.75 | 78.13 | 93.99 | 79.14 | 684.4 | **87.27** |
| **CSQA** | Zero-shot CoT | 93.35 | 93.40 | 64.61 | 65.24 | 98.87 | 67.29 | 87.07 | 91.50 | 77.94 | 412.3 | 83.83 |
| | Self-Refine | 84.66 | 94.21 | 66.89 | 60.92 | 96.87 | 63.87 | 87.11 | 87.35 | 74.55 | 379.4 | 77.61 |
| | Role-Playing | 83.21 | 94.33 | 66.03 | 59.93 | 96.76 | 60.86 | 85.91 | 88.23 | 73.85 | 350.6 | 76.20 |
| | *More think* | 97.02 | 91.16 | 68.96 | 64.32 | 98.99 | 62.00 | 80.90 | 92.38 | 77.17 | 786.1 | 85.85 |
| | Ours(+SFT) | 87.07 | 90.66 | 65.30 | 60.62 | 98.53 | 57.66 | 69.04 | 91.79 | 73.83 | **1003** | 83.10 |
| | Ours(+SFT+GRPO) | 87.98 | 91.34 | 65.41 | 61.59 | 98.74 | 61.28 | 70.00 | 92.00 | 77.86 | 824.6 | 86.85 |
| | Ours(+SFT+GRPO r.s.) | 92.12 | 92.59 | 67.86 | 66.00 | 98.82 | 70.38 | 79.26 | 91.10 | 79.75 | 684.6 | **87.96** |
| **GSM8K** | Zero-shot CoT | 68.51 | 93.85 | 76.33 | 55.83 | 94.35 | 35.73 | 81.47 | 68.51 | 65.05 | 236.3 | 68.08 |
| | Self-Refine | 80.59 | 93.90 | 67.67 | 65.78 | 98.32 | 61.71 | 83.78 | 84.69 | 75.55 | 352.3 | 80.37 |
| | Role-Playing | 74.00 | 93.29 | 77.62 | 59.76 | 95.40 | 41.27 | 80.61 | 73.40 | 68.53 | 292.1 | 72.87 |
| | *More think* | 89.02 | 92.41 | 78.25 | 61.33 | 98.44 | 64.84 | 79.81 | 83.89 | 74.61 | 564.4 | 81.79 |
| | Ours(+SFT) | 73.77 | 92.22 | 70.51 | 57.20 | 95.56 | 42.94 | 72.69 | 81.99 | 68.59 | **620.0** | 74.87 |
| | Ours(+SFT+GRPO) | 80.98 | 89.36 | 60.29 | 63.31 | 96.36 | 16.85 | 65.16 | 93.74 | 73.35 | 555.0 | 82.16 |
| | Ours(+SFT+GRPO r.s.) | 74.45 | 93.33 | 71.86 | 59.76 | 95.38 | 45.26 | 78.55 | 79.94 | 75.16 | 419.1 | **82.46** |

Table 12: Detailed composition of the diversity scores based on the output of R1-Distilled-Llama-8B. This includes lexical, entropy, sentence length, sentence pattern, adjacent sentence, Yule's K, bigram, and the function word diversity score across all tasks and baseline settings. In addition, we also provide the combined diversity score, average reasoning length and length normalized diversity score.

| R1-Distill-Llama-8B | | Diversity Sub Scores (in %) | | | | | | | | Combined | Len. | Norm |
|---|---|---|---|---|---|---|---|---|---|---|---|---|
| | | lex. | ent. | len. | pat. | adj. | yule. | bi. | func. | | | |
| **BBQ** | Zero-shot CoT | 73.65 | 94.29 | 70.10 | 66.72 | 96.75 | 44.49 | 84.99 | 88.07 | 74.58 | 236.5 | 79.92 |
| | Self-Refine | 91.04 | 92.36 | 75.58 | 57.25 | 98.89 | 54.38 | 77.76 | 89.80 | 73.03 | 236.5 | 75.85 |
| | Role-Playing | 81.70 | 94.06 | 78.18 | 69.67 | 97.38 | 52.19 | 84.85 | 88.66 | 77.49 | 347.1 | 80.91 |
| | *More think* | 90.33 | 91.74 | 67.77 | 61.75 | 98.69 | 46.29 | 79.61 | 92.18 | 74.11 | 533.8 | 84.11 |
| | Ours(+SFT) | 65.30 | 89.02 | 53.60 | 60.24 | 98.25 | 43.44 | 46.72 | 92.85 | 69.57 | **3353** | 82.64 |
| | Ours(+SFT+GRPO) | 84.02 | 90.87 | 55.86 | 61.07 | 98.62 | 52.50 | 62.01 | 93.41 | 75.91 | 949.7 | 85.75 |
| | Ours(+SFT+GRPO r.s.) | 91.44 | 92.51 | 67.30 | 66.00 | 99.52 | 68.55 | 73.76 | 93.10 | 81.38 | 673.6 | **89.58** |
| **GLOQA** | Zero-shot CoT | 96.46 | 93.43 | 61.57 | 60.66 | 98.75 | 82.35 | 87.29 | 90.88 | 77.49 | 571.5 | 87.07 |
| | Self-Refine | 90.19 | 95.70 | 60.93 | 62.19 | 98.93 | 84.22 | 92.21 | 89.12 | 77.97 | 244.3 | 81.11 |
| | Role-Playing | 94.77 | 94.21 | 66.43 | 60.15 | 98.70 | 83.01 | 88.66 | 90.05 | 77.45 | 453.0 | 83.02 |
| | *More think* | 99.45 | 91.38 | 65.28 | 59.25 | 99.23 | 82.57 | 83.31 | 92.50 | 77.35 | 991.4 | 87.19 |
| | Ours(+SFT) | 90.57 | 87.94 | 52.22 | 55.43 | 98.63 | 73.34 | 57.01 | 93.77 | 72.51 | **3852** | 87.26 |
| | Ours(+SFT+GRPO) | 99.20 | 89.37 | 52.52 | 56.55 | 98.54 | 75.67 | 69.56 | 93.51 | 77.10 | 1613.4 | 89.36 |
| | Ours(+SFT+GRPO r.s.) | 99.73 | 90.43 | 62.76 | 61.59 | 99.20 | 79.14 | 74.05 | 94.08 | 80.71 | 1225.6 | **91.78** |
| **CALI** | Zero-shot CoT | 77.26 | 93.88 | 71.84 | 59.48 | 97.44 | 35.53 | 85.60 | 90.57 | 71.63 | 246.0 | 73.98 |
| | Self-Refine | 84.48 | 91.97 | 77.03 | 65.29 | 98.13 | 27.89 | 81.28 | 91.02 | 73.66 | 458.7 | 78.87 |
| | Role-Playing | 82.94 | 92.26 | 74.30 | 63.07 | 96.97 | 27.42 | 82.03 | 91.90 | 72.62 | 392.5 | 77.32 |
| | *More think* | 93.95 | 90.62 | 74.81 | 57.13 | 98.41 | 39.30 | 77.90 | 93.49 | 72.19 | 683.1 | 80.30 |
| | Ours(+SFT) | 69.20 | 86.72 | 66.29 | 58.40 | 97.02 | 13.37 | 48.30 | 93.92 | 66.71 | **3279** | 79.77 |
| | Ours(+SFT+GRPO) | 84.56 | 88.53 | 65.38 | 61.54 | 96.78 | 20.42 | 62.40 | 94.13 | 73.28 | 1027.6 | 83.37 |
| | Ours(+SFT+GRPO r.s.) | 91.08 | 90.92 | 76.38 | 72.07 | 98.72 | 43.64 | 73.34 | 94.40 | 82.13 | 709.6 | **90.55** |
| **ETHICS** | Zero-shot CoT | 86.37 | 94.82 | 71.05 | 60.86 | 97.87 | 70.58 | 89.07 | 90.28 | 76.31 | 331.3 | 79.44 |
| | Self-Refine | 86.17 | 94.66 | 70.67 | 62.94 | 97.77 | 68.58 | 88.77 | 90.64 | 76.95 | 302.6 | 80.17 |
| | Role-Playing | 86.20 | 94.72 | 71.02 | 62.45 | 97.69 | 67.68 | 88.63 | 90.02 | 76.55 | 314.0 | 79.78 |
| | *More think* | 97.96 | 91.94 | 73.10 | 59.05 | 98.92 | 65.59 | 82.95 | 93.82 | 76.13 | 640.0 | 83.99 |
| | Ours(+SFT) | 85.48 | 90.17 | 61.54 | 59.99 | 97.89 | 40.09 | 69.97 | 94.78 | 72.11 | **1551.4** | 81.36 |
| | Ours(+SFT+GRPO) | 94.56 | 88.19 | 61.25 | 59.31 | 97.83 | 48.44 | 63.77 | 95.48 | 75.80 | 1364 | 87.89 |
| | Ours(+SFT+GRPO r.s.) | 95.06 | 91.68 | 78.70 | 77.34 | 99.27 | 73.82 | 77.40 | 94.82 | 88.22 | 805.3 | **96.54** |
| **CSQA** | Zero-shot CoT | 92.09 | 93.69 | 64.10 | 65.02 | 98.86 | 69.97 | 87.38 | 91.19 | 77.99 | 411.5 | 83.39 |
| | Self-Refine | 92.51 | 93.23 | 68.88 | 62.96 | 98.66 | 67.56 | 85.69 | 89.65 | 76.75 | 506.3 | 82.61 |
| | Role-Playing | 90.80 | 93.55 | 65.85 | 63.52 | 98.42 | 66.75 | 86.07 | 91.08 | 76.96 | 432.3 | 82.27 |
| | *More think* | 95.42 | 91.14 | 69.73 | 62.57 | 99.00 | 59.63 | 79.81 | 92.43 | 76.18 | 757.8 | 84.73 |
| | Ours(+SFT) | 67.44 | 87.55 | 61.90 | 55.26 | 98.87 | 51.86 | 44.03 | 92.22 | 68.67 | **4477** | 83.88 |
| | Ours(+SFT+GRPO) | 89.05 | 89.67 | 66.29 | 60.48 | 98.88 | 58.15 | 63.83 | 92.61 | 76.88 | 1271.7 | 87.96 |
| | Ours(+SFT+GRPO r.s.) | 94.03 | 91.01 | 73.49 | 69.59 | 99.48 | 68.95 | 72.66 | 92.98 | 83.28 | 989.0 | **92.98** |
| **GSM8K** | Zero-shot CoT | 75.15 | 92.69 | 79.99 | 63.63 | 96.13 | 43.06 | 78.06 | 77.24 | 71.11 | 394.8 | 76.52 |
| | Self-Refine | 82.66 | 92.90 | 70.87 | 64.75 | 98.32 | 59.52 | 80.61 | 85.40 | 75.12 | 490.5 | 81.24 |
| | Role-Playing | 76.89 | 93.01 | 79.21 | 61.12 | 95.94 | 45.48 | 79.57 | 74.53 | 69.93 | 347.1 | 75.02 |
| | *More think* | 90.63 | 91.13 | 79.86 | 61.87 | 98.65 | 65.24 | 76.14 | 86.37 | 75.30 | 830.4 | 84.12 |
| | Ours(+SFT) | 62.66 | 89.11 | 68.16 | 57.87 | 98.48 | 50.92 | 46.17 | 87.38 | 68.87 | **1961** | 81.53 |
| | Ours(+SFT+GRPO) | 74.74 | 90.22 | 74.48 | 60.17 | 98.42 | 56.18 | 58.18 | 87.15 | 74.85 | 1112.9 | 85.31 |
| | Ours(+SFT+GRPO r.s.) | 80.31 | 90.89 | 78.08 | 65.40 | 98.80 | 62.93 | 64.88 | 87.12 | 78.73 | 965.2 | **88.45** |

Table 13: Detailed composition of the diversity scores based on the output of R1-Distilled-Qwen-14B. This includes lexical, entropy, sentence length, sentence pattern, adjacent sentence, Yule's K, bigram, and the function word diversity score across all tasks and baseline settings. In addition, we also provide the combined diversity score, average reasoning length and normalized diversity score.

| R1-Distill-Qwen-14B | | Diversity Sub Scores (in %) | | | | | | | | Combined | Len. | Norm |
|---|---|---|---|---|---|---|---|---|---|---|---|---|
| | | lex. | ent. | len. | pat. | adj. | yule. | bi. | func. | | | |
| **BBQ** | Zero-shot CoT | 54.86 | 96.37 | 77.82 | 50.83 | 90.24 | 36.08 | 74.85 | 61.10 | 71.11 | 394.83 | 76.52 |
| | Self-Refine | 88.78 | 93.52 | 67.58 | 61.90 | 98.76 | 54.14 | 83.45 | 89.33 | 75.12 | 490.51 | 81.24 |
| | Role-Playing | 85.20 | 94.33 | 72.03 | 69.34 | 98.44 | 56.59 | 86.00 | 89.33 | 69.93 | 347.10 | 75.02 |
| | *More think* | 91.52 | 92.79 | 67.79 | 62.80 | 98.76 | 52.56 | 83.48 | 91.74 | 75.30 | 830.40 | 84.12 |
| | Ours(+SFT) | 87.11 | 91.63 | 55.95 | 56.50 | 93.06 | 54.77 | 69.39 | 89.92 | 70.57 | **793.83** | 78.09 |
| | Ours(+SFT+GRPO) | 96.25 | 93.30 | 58.72 | 62.31 | 98.00 | 73.61 | 82.21 | 92.02 | 80.13 | 552.4 | 86.88 |
| | Ours(+SFT+GRPO r.s.) | 97.18 | 94.31 | 70.63 | 68.30 | 98.77 | 80.12 | 87.35 | 91.70 | 84.44 | 407.4 | **90.17** |
| **GLOQA** | Zero-shot CoT | 83.87 | 95.02 | 52.24 | 55.62 | 93.18 | 77.30 | 87.18 | 86.53 | 72.52 | 383.8 | 73.28 |
| | Self-Refine | 61.11 | 98.17 | 53.82 | 35.76 | 78.45 | 71.26 | 86.63 | 67.96 | 57.85 | 77.07 | 49.77 |
| | Role-Playing | 78.21 | 95.95 | 52.35 | 48.33 | 87.36 | 72.05 | 88.46 | 79.14 | 66.85 | 252.4 | 65.55 |
| | *More think* | 99.25 | 93.20 | 64.47 | 55.58 | 99.15 | 84.44 | 86.46 | 90.58 | 75.88 | 606.1 | 83.57 |
| | Ours(+SFT) | 98.82 | 90.67 | 53.21 | 56.30 | 97.54 | 83.65 | 77.02 | 91.77 | 74.98 | **1304** | 85.98 |
| | Ours(+SFT+GRPO) | 99.20 | 91.97 | 56.63 | 61.23 | 98.38 | 86.28 | 82.16 | 92.07 | 80.93 | 923.0 | 90.33 |
| | Ours(+SFT+GRPO r.s.) | 99.12 | 93.58 | 65.23 | 65.87 | 98.69 | 89.05 | 87.05 | 91.65 | 84.00 | 598.5 | **91.32** |
| **CALI** | Zero-shot CoT | 59.85 | 95.73 | 72.54 | 54.47 | 93.75 | 27.88 | 83.20 | 84.21 | 66.36 | 102.1 | 58.51 |
| | Self-Refine | 87.48 | 92.47 | 73.54 | 65.19 | 97.80 | 28.50 | 83.81 | 88.92 | 73.36 | 355.2 | 77.86 |
| | Role-Playing | 74.21 | 93.17 | 61.54 | 53.45 | 85.33 | 25.36 | 83.40 | 78.82 | 63.83 | 252.6 | 65.06 |
| | *More think* | 87.17 | 91.43 | 72.84 | 61.01 | 98.54 | 24.83 | 80.47 | 91.74 | 71.69 | 436.6 | 77.87 |
| | Ours(+SFT) | 86.66 | 90.51 | 61.03 | 62.61 | 96.22 | 20.68 | 71.45 | 91.72 | 70.56 | **593.5** | 78.17 |
| | Ours(+SFT+GRPO) | 92.26 | 92.39 | 65.89 | 66.24 | 98.30 | 41.32 | 82.20 | 92.04 | 78.81 | 433.6 | 84.92 |
| | Ours(+SFT+GRPO r.s.) | 93.87 | 92.79 | 75.28 | 71.34 | 99.17 | 51.31 | 83.41 | 92.32 | 82.88 | 450.9 | **89.08** |
| **ETHICS** | Zero-shot CoT | 70.98 | 95.20 | 60.25 | 52.49 | 82.41 | 52.21 | 84.11 | 72.98 | 64.58 | 226.1 | 62.34 |
| | Self-Refine | 90.30 | 93.96 | 75.90 | 65.63 | 98.46 | 62.46 | 87.17 | 89.75 | 77.66 | 353.5 | 81.37 |
| | Role-Playing | 76.57 | 95.57 | 62.47 | 52.52 | 85.15 | 61.84 | 85.82 | 77.83 | 67.30 | 260.5 | 65.70 |
| | *More think* | 95.95 | 93.40 | 71.67 | 58.87 | 99.36 | 64.54 | 88.15 | 93.20 | 76.03 | 386.1 | 81.89 |
| | Ours(+SFT) | 82.15 | 93.70 | 49.78 | 46.43 | 77.94 | 53.32 | 81.54 | 76.66 | 62.39 | **472.5** | 63.99 |
| | Ours(+SFT+GRPO) | 97.79 | 93.68 | 68.78 | 65.92 | 99.41 | 75.80 | 87.94 | 93.03 | 83.20 | 438.9 | 89.42 |
| | Ours(+SFT+GRPO r.s.) | 97.74 | 94.43 | 75.08 | 72.86 | 99.72 | 81.62 | 89.69 | 92.88 | 87.35 | 369.8 | **92.89** |
| **CSQA** | Zero-shot CoT | 91.03 | 94.10 | 64.83 | 64.86 | 98.92 | 68.22 | 88.52 | 91.17 | 77.82 | 320.5 | 82.74 |
| | Self-Refine | 89.15 | 94.24 | 66.42 | 63.07 | 98.12 | 70.11 | 87.21 | 87.95 | 76.49 | 368.9 | 80.18 |
| | Role-Playing | 84.58 | 94.71 | 60.20 | 58.45 | 93.06 | 64.56 | 86.97 | 85.59 | 72.59 | 288.6 | 74.40 |
| | *More think* | 93.93 | 92.34 | 67.56 | 63.08 | 99.01 | 60.30 | 83.58 | 91.46 | 76.33 | 539.3 | 83.33 |
| | Ours(+SFT) | 91.32 | 91.03 | 62.71 | 57.27 | 93.64 | 64.58 | 70.18 | 90.39 | 72.57 | **1034** | 81.56 |
| | Ours(+SFT+GRPO) | 96.76 | 93.07 | 70.76 | 65.68 | 99.46 | 76.00 | 84.42 | 91.20 | 82.59 | 572.2 | 89.64 |
| | Ours(+SFT+GRPO r.s.) | 97.01 | 93.99 | 79.44 | 70.14 | 99.65 | 80.52 | 87.69 | 90.66 | 85.62 | 435.7 | **91.61** |
| **GSM8K** | Zero-shot CoT | 61.86 | 94.52 | 76.54 | 57.65 | 92.91 | 27.38 | 83.22 | 64.09 | 63.71 | 159.3 | 64.44 |
| | Self-Refine | 84.47 | 93.37 | 71.00 | 63.94 | 98.37 | 63.29 | 82.11 | 83.86 | 75.06 | 400.6 | 80.81 |
| | Role-Playing | 77.46 | 93.12 | 76.76 | 60.95 | 96.41 | 48.59 | 80.24 | 76.64 | 70.59 | 342.5 | 75.73 |
| | *More think* | 89.66 | 91.93 | 78.18 | 61.42 | 98.76 | 65.34 | 78.15 | 85.33 | 74.94 | 637.4 | 82.79 |
| | Ours(+SFT) | 77.96 | 90.69 | 69.04 | 59.71 | 98.38 | 61.19 | 61.77 | 87.23 | 72.26 | **1027** | 82.32 |
| | Ours(+SFT+GRPO) | 88.47 | 93.02 | 71.68 | 62.70 | 98.40 | 72.76 | 79.59 | 87.15 | 79.40 | 526.0 | 86.36 |
| | Ours(+SFT+GRPO r.s.) | 90.64 | 93.59 | 75.64 | 64.12 | 98.43 | 76.79 | 82.93 | 86.73 | 80.86 | 448.5 | **87.24** |

Table 14: Detailed composition of the diversity scores based on the output of Qwen3-8B. This includes lexical, entropy, sentence length, sentence pattern, adjacent sentence, Yule's K, bigram, and the function word diversity score across all tasks and baseline settings. In addition, we also provide the combined diversity score, average reasoning length and length normalized diversity score.

| Qwen3-8B | Diversity Sub Scores (in %) | | | | | | | | Combined | Len. | Norm |
|---|---|---|---|---|---|---|---|---|---|---|---|
| | lex. | ent. | len. | pat. | adj. | yule. | bi. | func. | | | |
| **BBQ** | | | | | | | | | | | |
| Zero-shot CoT | 52.47 | 95.16 | 75.87 | 54.75 | 97.52 | 38.52 | 80.28 | 67.14 | 64.01 | 75.09 | 54.22 |
| Self-Refine | 24.24 | 83.84 | 82.46 | 47.60 | 99.28 | 37.07 | 14.88 | 86.08 | 57.16 | 220.27 | 76.40 |
| Role-Playing | 21.33 | 58.13 | 82.45 | 36.31 | 70.59 | 31.03 | 14.56 | 66.24 | 43.76 | 364.63 | 61.64 |
| *More think* | 74.88 | 90.68 | 76.94 | 69.03 | 97.32 | 22.41 | 67.86 | 91.62 | 73.43 | 501.73 | 80.07 |
| Ours(+SFT) | 80.65 | 91.22 | 58.74 | 60.91 | 97.19 | 46.12 | 65.05 | 91.45 | 71.77 | **837.33** | 80.40 |
| Ours(+SFT+GRPO) | 87.37 | 93.30 | 59.98 | 65.35 | 98.47 | 58.11 | 79.68 | 91.66 | 79.41 | 490.2 | 85.47 |
| Ours(+SFT+GRPO r.s.) | 91.21 | 93.99 | 64.23 | 67.95 | 99.37 | 69.96 | 83.02 | 91.36 | 82.41 | 430.1 | **88.15** |
| **GLOQA** | | | | | | | | | | | |
| Zero-shot CoT | 60.44 | 92.40 | 38.83 | 28.31 | 67.34 | 50.23 | 80.00 | 61.78 | 49.02 | 195.91 | 46.72 |
| Self-Refine | 94.74 | 94.74 | 74.60 | 61.12 | 99.73 | 80.21 | 87.87 | 87.73 | 77.58 | 345.58 | 82.54 |
| Role-Playing | 51.70 | 49.38 | 96.07 | 59.98 | 89.77 | 52.69 | 28.89 | 75.49 | 59.64 | 515.61 | 73.72 |
| *More think* | 94.98 | 90.68 | 63.19 | 59.97 | 98.22 | 53.66 | 75.22 | 93.42 | 74.06 | 824.96 | 83.04 |
| Ours(+SFT) | 90.48 | 90.17 | 54.69 | 56.89 | 97.49 | 71.19 | 66.16 | 91.74 | 73.05 | **2114.03** | 84.40 |
| Ours(+SFT+GRPO) | 94.40 | 91.63 | 57.27 | 59.96 | 98.09 | 74.66 | 74.66 | 92.14 | 78.58 | 972.1 | 87.74 |
| Ours(+SFT+GRPO r.s.) | 97.51 | 92.08 | 59.33 | 62.39 | 99.15 | 79.93 | 77.83 | 92.47 | 80.82 | 901.9 | **89.88** |
| **CALI** | | | | | | | | | | | |
| Zero-shot CoT | 47.20 | 96.84 | 43.57 | 32.92 | 65.04 | 16.21 | 82.47 | 60.30 | 46.86 | 65.29 | 41.34 |
| Self-Refine | 45.19 | 51.17 | 96.50 | 43.83 | 60.72 | 1.39 | 25.82 | 73.98 | 49.47 | 369.00 | 64.83 |
| Role-Playing | 38.58 | 50.86 | 91.20 | 55.32 | 85.93 | 0.52 | 22.89 | 75.85 | 56.12 | 331.59 | 72.10 |
| *More think* | 65.61 | 89.47 | 67.08 | 63.35 | 93.27 | 7.80 | 64.55 | 91.73 | 68.13 | 923.11 | 74.64 |
| Ours(+SFT) | 81.37 | 90.45 | 64.59 | 63.80 | 96.12 | 12.97 | 72.05 | 92.65 | 70.38 | **1572.03** | 77.38 |
| Ours(+SFT+GRPO) | 83.56 | 91.77 | 63.57 | 65.36 | 96.50 | 17.32 | 80.39 | 91.89 | 75.14 | 348.2 | 80.36 |
| Ours(+SFT+GRPO r.s.) | 86.61 | 92.88 | 65.76 | 68.83 | 97.70 | 33.11 | 84.24 | 92.31 | 78.99 | 316.3 | **83.84** |
| **ETHICS** | | | | | | | | | | | |
| Zero-shot CoT | 79.75 | 95.10 | 42.62 | 62.07 | 99.88 | 36.57 | 97.33 | 86.35 | 71.48 | 136.00 | 72.87 |
| Self-Refine | 41.18 | 56.57 | 95.91 | 45.36 | 70.87 | 2.84 | 23.91 | 75.89 | 51.57 | 206.86 | 67.68 |
| Role-Playing | 36.58 | 55.54 | 91.01 | 44.59 | 72.84 | 4.11 | 19.63 | 75.79 | 50.83 | 231.06 | 68.46 |
| *More think* | 72.72 | 90.15 | 76.00 | 62.81 | 96.46 | 18.46 | 67.57 | 93.68 | 70.68 | 536.94 | 77.68 |
| Ours(+SFT) | 100.00 | 92.19 | 47.93 | 54.97 | 97.40 | 61.03 | 87.14 | 92.23 | 72.64 | 596.00 | 80.68 |
| Ours(+SFT+GRPO) | 89.29 | 91.17 | 68.53 | 63.01 | 97.76 | 52.62 | 73.12 | 94.18 | 78.33 | **860.3** | 86.84 |
| Ours(+SFT+GRPO r.s.) | 93.28 | 92.45 | 70.57 | 66.67 | 98.81 | 63.73 | 79.87 | 93.49 | 81.76 | 637.2 | **89.09** |
| **CSQA** | | | | | | | | | | | |
| Zero-shot CoT | 57.42 | 83.79 | 42.16 | 45.30 | 64.68 | 29.57 | 67.74 | 61.30 | 52.34 | 243.99 | 53.18 |
| Self-Refine | 47.38 | 56.46 | 91.39 | 46.53 | 66.61 | 10.76 | 26.14 | 76.33 | 52.68 | 301.89 | 68.46 |
| Role-Playing | 23.52 | 73.73 | 54.89 | 54.14 | 86.12 | 9.12 | 14.84 | 82.24 | 55.98 | 324.66 | 75.27 |
| *More think* | 72.43 | 90.93 | 67.35 | 68.04 | 97.05 | 21.82 | 70.55 | 91.56 | 72.48 | 449.76 | 78.55 |
| Ours(+SFT) | 77.28 | 89.49 | 66.57 | 58.76 | 97.40 | 52.48 | 59.55 | 90.90 | 71.32 | **2362.13** | 82.54 |
| Ours(+SFT+GRPO) | 89.99 | 92.58 | 73.80 | 62.70 | 98.38 | 64.54 | 77.85 | 90.16 | 79.25 | 635.4 | 86.40 |
| Ours(+SFT+GRPO r.s.) | 91.83 | 92.64 | 68.36 | 64.45 | 98.93 | 68.67 | 77.92 | 90.46 | 80.39 | 652.0 | **87.93** |
| **GSM8K** | | | | | | | | | | | |
| Zero-shot CoT | 63.06 | 94.31 | 72.52 | 51.98 | 93.33 | 28.50 | 82.67 | 72.50 | 63.10 | 197.31 | 64.71 |
| Self-Refine | 89.59 | 92.80 | 78.47 | 69.80 | 98.78 | 60.36 | 80.10 | 87.05 | 78.29 | 522.58 | 84.82 |
| Role-Playing | 90.01 | 92.06 | 79.70 | 69.98 | 98.24 | 61.37 | 78.24 | 84.58 | 77.87 | 609.06 | 85.23 |
| *More think* | 76.08 | 90.70 | 79.40 | 66.39 | 95.59 | 36.75 | 69.22 | 85.40 | 72.64 | 664.80 | 80.09 |
| Ours(+SFT) | 64.43 | 89.74 | 74.44 | 61.24 | 97.39 | 45.65 | 50.38 | 86.10 | 69.97 | **1557.56** | 81.21 |
| Ours(+SFT+GRPO) | 75.49 | 91.48 | 78.45 | 65.40 | 98.15 | 54.89 | 66.01 | 85.58 | 77.42 | 922.7 | 85.98 |
| Ours(+SFT+GRPO r.s.) | 81.83 | 93.22 | 81.60 | 67.97 | 98.85 | 65.21 | 78.03 | 84.87 | 80.70 | 487.8 | **86.93** |

