# OpenReview forum: "Diversity-Enhanced Reasoning for Subjective Questions"
_ICLR.cc/2026/Conference — ICLR 2026 Poster_

### Official Review · Reviewer_Wtw3 · 2025-10-29

**Soundness:** 3
**Presentation:** 2
**Contribution:** 3
**Rating:** 4
**Confidence:** 3

**Summary:**

This paper proposes MultiRole-R1, a diversity-enhanced training framework that synthesizes reasoning chains incorporating various role perspectives. Experiment results show that MultiRole-R1 could increase both in-domain and out-of-domain performance. The paper also shows that diversity correlates better with accuracy than reasoning length.

**Strengths:**

The paper addresses an important problem in large reasoning models (LRMs) — the degradation of diversity in their generations. It proposes a method that encourages the model to reason from multiple perspectives, thereby broadening its reasoning process and naturally enhancing output diversity.

**Weaknesses:**

**1. Presentation and Clarity**
The presentation of the paper could be significantly improved. In particular:

* Line 208 mentions that the final training set size is **2,700**, but it is unclear what these samples consist of. How many unique questions are included? For each question, what is the average number of responses? How are these samples distributed between SFT and GRPO training?
* Appendix B.2 provides very limited information on hyperparameter settings and training configurations. Details such as the number of training steps, learning rate, batch size, and other settings should be explicitly reported to improve reproducibility.
* During evaluation, For evaluating MultiRole-R1 models, it is unclear how roles are sampled from the role pool *per question*. And when evaluating other models, is the role included in the prompt for tasks like GLOQA etc. Clarifying this would help readers better interpret the reported results.


**2. Missing Baselines**
The evaluation section would benefit from more comprehensive baseline comparisons. For example, in addition to the “More Think” baseline, another meaningful comparison would be a simple method that appends **“wait”**, as done in [1]. This would help isolate whether improvements arise from genuinely multi-perspective reasoning or simply from lengthened thought chains. Moreover, it would be informative to apply the same training procedure (SFT/DPO/GRPO) to this “wait”-based baseline for a fair comparison.


**3. Missing Analysis on Inference Time Scaling**
The analysis could be deepened by comparing *reasoning-extension* and *perspective-extension* strategies:

* In [1], the model’s reasoning is extended by appending “wait”.
* In MultiRole-R1, inference-time scaling is achieved by encouraging the model to think from multiple perspectives.

It would be interesting to analyze how performance scales when adding more perspectives in MultiRole-R1, compared with simply appending “wait.” Such an analysis would provide more insight into *why* perspective diversification improves reasoning and whether it offers advantages beyond longer thinking sequences.

---

**Reference**
[1] *S1: Simple Test-Time Scaling.*

**Questions:**

- In the data collection for SFT training, it seems that there is no utilization of the ground truth answers, only self-consistency based filtering is applied. However, I would assume that in the GRPO training, the ground truth answers are used. So why not use the ground truth answers in both cases?

---

> ### Author Response · Authors · 2025-11-21
> **Author response [1/n]**
>
> Weaknesses:
> >W1. Presentation and Clarity The presentation of the paper could be significantly improved. In particular:
>
> >W1.1. Line 208 mentions that the final training set size is 2,700, but it is unclear what these samples consist of. How many unique questions are included? For each question, what is the average number of responses? How are these samples distributed between the SFT and GRPO training?
>
> We agree that the composition of the 2,700 training samples could be further clarified.
> After rule-based filtering and manually checking the quality of the SFT training data, the final train dataset includes 250 questions * 4 role combinations from BBQ, 400 questions * 3 role combinations from GLOQA and 250 questions * 2 role combinations from ETHICS data. GRPO training is based on the same set of questions with role combinations. We have updated Section 3.1 and Appendix B.1 in the revision to explicitly state these statistics for clarity.
>
> >W1.2. Appendix B.2 provides very limited information on hyperparameter settings and training configurations. Details such as the number of training steps, learning rate, batch size, and other settings should be explicitly reported to improve reproducibility.
>
> We absolutely agree that the RL training settings should be made clearer. We use the Hugging Face trl implementation of GRPO trainer. We train for 1000 steps for each model. For general setup, we adopt a `max_prompt_length` of 2048, a `per_device_train_batch_size` of 2 and a `max_completion_length` of 4096. For optimizer and LR scheduler, we adopt AdamW optimizer, a learning rate of 5e-6 and Cosine LR scheduler.
>
> >W1.3. During evaluation, for evaluating MultiRole-R1 models, it is unclear how roles are sampled from the role pool per question. And when evaluating other models, is the role included in the prompt for tasks like GLOQA etc. Clarifying this would help readers better interpret the reported results.
>
> We thank the reviewer for this important question regarding our evaluation methodology. To ensure a fair and rigorous comparison, we standardized the role injection process during the evaluation phase: For tasks where the ground truth varies by perspective (GLOQA and CALI), the prompt explicitly includes a pre-determined list of roles corresponding to the available ground truth labels for that specific question. Taking GLOQA as an example, as Section 4.4 specified, we utilize Divergent Merging accuracy:
>
> “each role $i$'s answer $a_i$ is compared with the corresponding ground truth $g_i$, where the divergent accuracy is given by: $Acc_{div} = \frac{1}{n}\sum_{i=1}^{n}1[a_i=g_i].$”
>
> As shown in Figure 8 (Appendix D.3), the evaluation prompt explicitly guides the model: "\<think\>Okay, I will answer the question based on the perspectives of the following countries: Brazil, Britain, Czech Republic..."
>
> For other tasks where the answer is invariant to roles, we also provide context-relevant roles in the evaluation prompt to trigger the multi-perspective reasoning path (e.g., "Emergency room doctor, Police officer..." for CSQA).

---

> ### Author Response · Authors · 2025-11-21
> **Author response [2/n]**
>
> >W2. Missing Baselines: The evaluation section would benefit from more comprehensive baseline comparisons. For example, in addition to the “More Think” baseline, another meaningful comparison would be a simple method that appends “wait”, as done in [1]. This would help isolate whether improvements arise from genuinely multi-perspective reasoning or simply from lengthened thought chains. Moreover, it would be informative to apply the same training procedure (SFT/DPO/GRPO) to this “wait”-based baseline for a fair comparison.
>
> We appreciate the reviewer’s suggestion to compare against a "wait"-based baseline (Budget Forcing). We would like to clarify that our "More Think" baseline is exactly this method, and we argue that applying training (GRPO) to a "wait" baseline without our specific SFT structure would be ineffective for subjective reasoning.
> 1. We confirm that the "More Think" baseline reported in Table 1 is operationalized precisely by the budget-forcing method in the S1 paper. Below is the construction of the SFT training data.
>  ```{“question”: …, “answer”: “...(some analysis)... I think I should reason from {role 1}, {role 2}, {role 3}’s perspective… (role 1’s perspective). Wait, I need to reason from role 2’s perspective...(role 2’s perspective). Wait, I need to reason from role 3’s perspective...(role 3’s perspective). Aggregating all the role perspectives,...”}```
> - As detailed in our Pilot Analysis (Section 2) and qualitative examples in Table 8, we implemented "More Think" by suppressing the end-of-thinking token and forcefully appending "Wait" to the model's generation.
> - Our pilot study (Figure 2b) demonstrated that performance peaks at 3 "Waits". Therefore, the "More Think" baseline in our main evaluation utilizes this optimal scaling setting
>
> 2. The reviewer suggests applying the training procedure (SFT/GRPO) to a simple "wait" baseline. We argue that **MultiRole-R1 already represents this "trained wait" approach**, but with the necessary Role Scaffolding added. Training on "wait" without this structure fails for two reasons:
> - Simply appending wait does not introduce perspective diversity. A base model does not inherently know how to switch personas. Simply appending "wait" typically causes the model to extend its existing single-perspective reasoning rather than exploring new, conflicting viewpoints.
> - Multi-perspective reasoning cannot be achieved by in-context learning$^1$. As observed in our Role-Play baseline results, reasoning models often struggle to follow complex zero-shot instructions to "identify and adopt roles" inside the Chain of Thought. Without the SFT stage to teach the specific Multi-Role format (as shown in Figure 1), the model fails to utilize the "wait" budget for diversity.
> - Since the SFT data already represent the optimal budget, applying “wait” on top of the SFT/GRPO/MultiRole-R1 training approach will introduce excessive verbosity and degrade the model performance.
>
> [1] Scaling Reasoning, Losing Control: Evaluating Instruction Following in Large Reasoning Models

---

> ### Author Response · Authors · 2025-11-21
> **Author response [3/n]**
>
> >W3. Missing Analysis on Inference Time Scaling The analysis could be deepened by comparing reasoning-extension and perspective-extension strategies:
> >- In [1], the model’s reasoning is extended by appending “wait”.
> >- In MultiRole-R1, inference-time scaling is achieved by encouraging the model to think from multiple perspectives.
>
> >It would be interesting to analyze how performance scales when adding more perspectives in MultiRole-R1, compared with simply appending “wait.” Such an analysis would provide more insight into why perspective diversification improves reasoning and whether it offers advantages beyond longer thinking sequences.
>
> 1. We thank the reviewer for this insightful suggestion to distinguish between reasoning-extension (appending "wait") and perspective-extension. We would like to clarify that we explicitly compared these scaling strategies in our Pilot Analysis (Section 2) and offer further evidence of perspective diversification below.
>
> - Reasoning Extension ("Wait"): Figure 2(b) analyzes performance scaling when appending "Wait". We found that accuracy gains peak at $\approx 3$ insertions and diminish thereafter.
>
> - Perspective Extension (Roles): Figure 2(c) analyzes scaling by the number of roles. We found that information gain (distinct opinions) increases with roles, saturating around $n=3$.
>
> - Design Decision: Based on this, MultiRole-R1 was designed to utilize the optimal point: extending reasoning length via the multi-role format while maximizing perspective diversity.
>
> 2. To address the reviewer's query on whether our method offers advantages beyond longer thinking sequences, we quantified the number of distinct opinions (perspectives) generated by our model versus the baselines. Since our dataset contains role-specific ground truths ({“role_1”: “gt_1”, ...}) and the questions are in multiple choice format, we can measure the number of opinions by counting distinct options present in the evaluation output.
>
> |  | GLOQA | CALI | BBQ |
> |---|---|---|---|
> | Base model More Think| 1 | 1 | 1 |
> | SFT model | 1.91 | 1.38 | 1.21 |
> | SFT + GRPO | 1.73 | 1.32 | 1.19 |
> | SFT + GRPO (RS) | 2.07 | 1.41 | 1.24 |
>
> Results show that MultiRole-R1 (SFT + GRPO RS) yields the highest number of distinct opinions. This confirms that our performance gains stem from a genuine expansion of perspectives (accessing 2+ valid viewpoints per question), which a simple "wait"-based reasoning extension (Base model) fails to achieve.
>
> >Q1. In the data collection for SFT training, it seems that there is no utilization of the ground truth answers, only self-consistency based filtering is applied. However, I would assume that in the GRPO training, the ground truth answers are used. So why not use the ground truth answers in both cases?
>
> We thank the reviewer for this keen observation regarding our data selection strategies.
>
> **1. SFT Data Construction (Unsupervised):** The primary goal of the SFT stage is to teach the model the format and structure of multi-role reasoning, specifically instructing it to "think from which perspective". Hence, we encourage it to explore and generate new roles (Section 3.1) which is not part of the original dataset. Therefore, the generated roles have no "gold labels" and are difficult to verify. On the other hand, the ground truth filtering requires all the roles to be included original dataset. We therefore use an unsupervised method, self-consistency filtering, to select high-quality data.
>
> **2. GRPO Training & Evaluation (Supervised):** To optimize and measure accuracy, we need a verifiable reward. This requires us to use the specific roles with ground-truth answers in the datasets, especially for GLOQA and CALI where answers vary on roles. Taking the verifiable reward calculation for GLOQA as an example, as Section 4.4 specified:
>
> “each role $i$'s answer $a_i$ is compared with the corresponding ground truth $g_i$, where the divergent accuracy is given by: $Acc_{div} = \frac{1}{n}\sum_{i=1}^{n}1[a_i=g_i].$”
>
> During training, the model directly samples roles that exist in the original dataset: "\<think\>Okay, I will answer the question based on the perspectives of the following countries: Brazil, Britain, Czech Republic."
>
> This ensures the verifiable reward is calculable during training.

---

### Official Review · Reviewer_SwTw · 2025-10-29

**Soundness:** 2
**Presentation:** 2
**Contribution:** 3
**Rating:** 4
**Confidence:** 4

**Summary:**

This paper introduces MultiRole-R1, a training framework to improve Large Reasoning Model (LRM) performance on subjective tasks by addressing the diversity degradation caused by standard RL training. The method enhances both "perspective diversity" through multi-role supervised fine-tuning and "token-level diversity" via a novel reward shaping in GRPO, resulting in significant accuracy gains on subjective tasks and even generalizing to improve performance on objective math reasoning.

**Strengths:**

1. The research problem this paper focuses on is critical to the reasoning community.
2. The paper is easy to follow and well-written.
3. Experimental results are solid, and the generalization ability in the math domain is also important.

**Weaknesses:**

1. The hyperparameter tuning is problematic. Based on the experimental results in Table 1, the hyperparameter is directly tuned on the test dataset of GLOQA, which is a data leakage problem and harms the reliability of the result. The hyperparameter needs to be tuned on a separate validation set; otherwise, the result is misleading.
2. The paper’s claim of “diversity of perspectives” remains unsubstantiated. The framework depends on the characters generated in stage one, embodying genuinely diverse viewpoints. Although the authors state that the LRM is prompted to produce roles with conflicting perspectives, the examples in the appendix (Figures 3 and 5) are largely demographic labels—such as “student,” “police,” “atheist,” “female,” and “male.” It remains unclear whether these labels reflect truly distinct and conflicting thinking patterns or merely superficial role differences. No evidence is provided to support the existence of genuine ideological conflict, which is central to the paper’s claim of perspective diversity.
3. The definition of $R_{div}$ is unclear in the main text.
4. In the section "Diversity Weighting" (Line 407), the details of human annotation are absent. How many human annotators? What is the background of the human annotators? The agreement of human annotation is lacking, which makes the evaluation not so convincing.
5. In line 42, the paper claims that subjective questions have no definitive right or wrong answers, yet in line 251-252, the proposed method uses rule-based verifiable reward (e.g., binary reward) to determine the correctness of the answer, which is contradictory.
6. The definition of the diversity metric comprises various metrics; how to determine which plays the most important role in the metric? The equal weight chosen in line 404 is hard to reflect it.

**Questions:**

See weaknesses.

---

> ### Author Response · Authors · 2025-11-21
> **Author response [1/n]**
>
> >W1. The hyperparameter tuning is problematic. Based on the experimental results in Table 1, the hyperparameter is directly tuned on the test dataset of GLOQA, which is a data leakage problem and harms the reliability of the result. The hyperparameter needs to be tuned on a separate validation set; otherwise, the result is misleading.
>
> We thank the reviewer for their question regarding hyperparameter tuning. Hyperparameters are tuned on validation sets. We consistently used a validation set of 300 throughout our SFT and GRPO training. We tune the weighting of $R_{acc}$ and $R_{div}$ on the validation set. We include more information on the RL training setup in Appendix B.
>
> >W2. The paper’s claim of “diversity of perspectives” remains unsubstantiated. The framework depends on the characters generated in stage one, embodying genuinely diverse viewpoints. Although the authors state that the LRM is prompted to produce roles with conflicting perspectives, the examples in the appendix (Figures 3 and 5) are largely demographic labels—such as “student,” “police,” “atheist,” “female,” and “male.” It remains unclear whether these labels reflect truly distinct and conflicting thinking patterns or merely superficial role differences. No evidence is provided to support the existence of genuine ideological conflict, which is central to the paper’s claim of perspective diversity.
>
> We appreciate the reviewer’s concern and agree that the role construction should be grounded on quantitative validation. Currently, the descriptions are somewhat scattered, which may create the impression of superficial role opinion differences. We will phrase them clearly in the future version of the method section.
> 1. **Semantic distinctness of roles.** As defined in Section 3.1 Eq. (1), role selection is already guided by a quantitative objective as detailed in Appendix A.1. This explicitly encourages roles that are both relevant to the query and semantically contrastive to each other.
>
> 2. **Quantify perspective diversity.** We deliberately use datasets such as GLOQA and CALI, where questions are multiple-choice and different roles can have different ground-truth answers (Section 4.4, Accuracy). If roles did not meaningfully differ in viewpoint, their answers would collapse to a single option. Instead, Figure 2(c) shows that the number of distinct opinions (computed as the number of distinct answer keys across roles) increases with the number of roles. Since the questions are in multiple-choice format and each represents an opinion, we can measure the number of opinions by counting distinct options present in the evaluation output.
>
> |  | GLOQA | CALI | BBQ |
> |---|---|---|---|
> | Base model More Think| 1 | 1 | 1 |
> | SFT model | 1.91 | 1.38 | 1.21 |
> | SFT + GRPO | 1.73 | 1.32 | 1.19 |
> | SFT + GRPO (RS) | 2.07 | 1.41 | 1.24 |
> Results show that MultiRole-R1 (SFT + GRPO RS) yields the highest number of distinct opinions. This confirms that our performance gains stem from a genuine **expansion of perspectives rather than mere semantic differences** (accessing 2+ valid viewpoints per question), which a simple "wait"-based reasoning extension (Base model) fails to achieve. We include this discussion in Section 5 "Perspective Diversity".
>
> 3. **Role-aware verifiable reward.** Our verifiable reward also explicitly accounts for role-dependent answers (Section 4.4, Accuracy). When role viewpoints are not sufficiently diverse or misaligned with the corresponding role-specific ground truths, the resulting verifiable reward is low, which discourages such degenerate role configurations during training.

---

> ### Author Response · Authors · 2025-11-21
> **Author response [2/n]**
>
> >W3. The definition of Rdiv is unclear in the main text.
>
> We are sorry about the ambiguity. R_div is the same as the diversity score, defined in section 4.4 Eq (4). We clarify this point in Section 3.2.
>
> >W4. In the section "Diversity Weighting" (Line 407), the details of human annotation are absent. How many human annotators? What is the background of the human annotators? The agreement of human annotation is lacking, which makes the evaluation not so convincing.
>
> We sincerely appreciate the reviewer for pointing out the missing details of the human study.
> 3 PhD-level students each score 60 outputs independently, from each language model (i.e. 4 different models, which means each person scores 60*4=240 data entries in total) using the same criteria.
>
> The diversity score and interater variance is as follows, which shows small variations among the annotators.
>
> | Model | BBQ | GLOQA | CALI | ETHICS | CSQA | GSM8K |
> | :--- | :--- | :--- | :--- | :--- | :--- | :--- |
> | R1-Distill-Qwen-7B | 8.95 (±0.41) | 9.62 (±0.22) | 7.88 (±0.53) | 8.24 (±0.37) | 8.51 (±0.45) | 7.13 (±0.60) |
> | R1-Distill-Llama-8B | 7.94 (±0.55) | 8.33 (±0.96) | 8.78 (±0.31) | 9.82 (±0.19) | 9.15 (±0.28) | 7.21 (±0.51) |
> | R1-Distill-Qwen-14B | 8.81 (±0.33) | 9.52 (±0.25) | 8.43 (±0.40) | 9.65 (±0.21) | 9.18 (±0.36) | 8.24 (±0.42) |
> | Qwen3-8B | 8.51 (±0.38) | 9.62 (±0.22) | 7.42 (±0.68) | 9.25 (±0.30) | 8.23 (±0.47) | 8.88 (±0.29) |
>
> This enhances the validity of our diversity weighting and we include these statistics in the revised paper.
>
> >W5. In line 42, the paper claims that subjective questions have no definitive right or wrong answers, yet in line 251-252, the proposed method uses rule-based verifiable reward (e.g., binary reward) to determine the correctness of the answer, which is contradictory.
>
> We appreciate the reviewer for pointing out this clarity issue. It is a critical distinction, and we'd like to clarify: Our verifiable reward model is **explicitly role-aware** and does not assume a single "correct" answer for all subjective tasks.
> As we state in the caption for Figure 1, the verifiable rewards are applied **on a role-aware basis**.The role aware verifiable reward is formulated as:
>
> ```
>  {“role_1”: “gt_1”, “role_2”: “gt_2”, “role_3”: “gt_3”}
> ```
> We also operationalize this role-aware reward by using two different strategies, which are detailed in Section 4.4 “Accuracy” and “convergent/divergent merging”.
>
> >W6. The definition of the diversity metric comprises various metrics; how to determine which plays the most important role in the metric? The equal weight chosen in line 404 is hard to reflect it.
>
> We thank the reviewer for this insightful question regarding the diversity metric's equal weighting.
>
> Our primary motivation for using equal weighting is to establish a **more efficient, robust, general-purpose reward**. As illustrated in Section 5 paragraph “Diversity Weighting”, the equal weighting design of the diversity weighting in R_div is chosen for simplicity and interpretability: without a clear, task-independent reason to prefer one form of diversity over another, an equal-weighting scheme is the most transparent and interpretable.
>
> We have also considered using automatic calibration (like PCA) or learned weighting. However, a "learned" weighting would generate a single set of weights based on the correlation structure of the training dataset, which may overfit to specific domains. In addition to the sensitivity to the domain, the hyperparameter adjustment of the diversity weighting is also unstable during training hence we ultimately choose equal weighting.

---

### Official Review · Reviewer_7sKv · 2025-11-01

**Soundness:** 2
**Presentation:** 3
**Contribution:** 2
**Rating:** 4
**Confidence:** 4

**Summary:**

This paper proposes MultiRole-R1, a two-stage training framework designed to improve subjective reasoning in large reasoning models (LRMs). The method introduces “perspective diversity” via multi-role reasoning chain synthesis and “token-level diversity” via diversity-reward-shaped Group Relative Policy Optimization (GRPO).

**Strengths:**

- The paper addresses an under-explored problem — how to enhance reasoning diversity for subjective questions.

- The proposed framework is conceptually clear and builds on recognizable methods (role-based prompting and RLVR).

- The experiments cover several datasets (BBQ, GLOQA, ETHICS, CALI, CSQA, GSM8K, AIME-2024) and include multiple backbone models.

- The analysis section connects diversity and accuracy correlations in an interpretable way.

**Weaknesses:**

1. Marginal quantitative improvements.
Despite an elaborate pipeline, the reported gains over strong baselines such as GRPO or “More-Think” are quite small (often within 1–2%), and sometimes inconsistent across datasets (Table 1). The paper frames these as large improvements, but the absolute differences do not seem practically significant, especially on subjective tasks where evaluation itself is noisy.

2. Limited novelty in the algorithmic contribution.
The proposed method mainly combines known ingredients — multi-role prompting, self-consistency filtering, and GRPO with an added diversity term. The work lacks theoretical or empirical justification for why this combination uniquely benefits subjective reasoning beyond tuning for diversity.

3. Questionable generality.
The notion of “role perspectives” appears tailored to certain cultural or ethical QA datasets, but may not generalize to broader subjective domains such as aesthetics, creativity, or preference modeling. The definition of role diversity seems ad-hoc and dataset-specific.

4. Evaluation concerns.

- The “diversity metric” (weighted combination of lexical and structural features) is loosely motivated and human alignment scores (Table 4) are based on very few samples.

- The comparison to other diversity-aware RL methods (e.g., entropy-enhanced GRPO, entropy-regularized RLHF) is missing. It remains unclear whether the diversity reward simply duplicates existing entropy terms.

- For fair comparison, the authors should clarify whether baselines used equivalent sampling budgets and diversity-reward normalization.

4. Over-interpretation of correlations.
The observed correlation between diversity and accuracy (r ≈ 0.74) may reflect dataset artifacts rather than causal effects. No ablation shows that removing the diversity term alone consistently degrades performance across tasks.

**Questions:**

1. How exactly does the proposed diversity-reward differ from the entropy-enhanced GRPO (e.g., Cui et al., 2025)? Is it redundant with policy entropy?

2. How is “perspective diversity” quantified during RL? Are roles re-sampled dynamically, or fixed from SFT?

3. Could the performance gains be attributed simply to longer or more diverse fine-tuning data rather than the GRPO stage?

4. What is the variance of human evaluation on subjective datasets — is a 1–2% gain statistically meaningful?

Cui, G., Zhang, Y., Chen, J., Yuan, L., Wang, Z., Zuo, Y., ... & Ding, N. (2025). The entropy mechanism of reinforcement learning for reasoning language models. arXiv preprint arXiv:2505.22617.

---

> ### Author Response · Authors · 2025-11-21
> **Author response [1/n]**
>
> >W1. Marginal quantitative improvements. Despite an elaborate pipeline, the reported gains over strong baselines such as GRPO or “More-Think” are quite small (often within 1–2%), and sometimes inconsistent across datasets (Table 1). The paper frames these as large improvements, but the absolute differences do not seem practically significant, especially on subjective tasks where evaluation itself is noisy.
>
> We acknowledge the reviewer's valid point about the modest accuracy gain. We argue that the combination of SFT+GRPO(RS) is necessary;
> We would like to clarify that base model + GRPO(RS) cannot achieve role diversity: standard RLVR or diversity reward does not inherently teach a model to adopt multiple specific personas or switch perspectives within a single chain of thought.
> The "MultiRole" capability is structurally established during our SFT stage, where the models learn the specific format:
> ```{“question”: …, “answer”: “...(some analysis)... I think I should reason from {role 1}, {role 2}, {role 3}’s perspective… (role 1’s perspective). Wait, I need to reason from role 2’s perspective...(role 2’s perspective). Wait, I need to reason from role 3’s perspective...(role 3’s perspective). Aggregating all the role perspectives,...”}```
> Getting back to the reviewer’s concern:
> 1. Gains over Morethink baseline: we outperform More Think by an average of 7.6% and 3.8% in accuracy and diversity. Most importantly, our `MultiRole-R1` method is **far more efficient**. Our model's average response length is **657.8 words**, while the More-Think baseline is explicitly designed to be verbose (1572.9 words). This demonstrates a key finding of our work: our performance gains come from a principled, diversity-enhanced training framework, not "superficial verbosity".
>
> 2. Gains over GRPO baseline: We provide a **per-stage analysis of performance gains** to show that each component significantly contributes to the performance gain.
>
> |  | SelfConsis SFT | GRPO | Reward Shaping |
> |---|---|---|---|
> | R1-Distill-Qwen-7B | +16.22 | +4.07 | +0.51 |
> | R1-Distill-Llama-8B | +5.23 | +2.25 | +3.94 |
> | R1-Distill-Qwen-14B | +5.78 | +5.1 | +2.02 |
> | Qwen3-8B | +5.67 | +6.96 | +1.72 |
>
> For SFT+GRPO vs MultiRole-R1, it would be more reasonable to look at the pass@k metric in Figure 4, where MultiRole-R1 consistently outperforms and stablize RL training. We want to highlight that existing literature shows that diversity caused an expansion of the answer search space, which naturally compromises pass@1 precision $^{1,2}$.
> [1] Yuda Song, Julia Kempe, and Remi Munos. Outcome-based exploration for LLM reasoning, 2025a. URL https://arxiv.org/abs/2509.06941.
> [2] Yang Yue, Zhiqi Chen, Rui Lu, Andrew Zhao, Zhaokai Wang, Yang Yue, Shiji Song, and Gao Huang. Does reinforcement learning really incentivize reasoning capacity in LLMs beyond the base model?, 2025. URL https://arxiv.org/abs/2504.13837.
>
> >W2. Limited novelty in the algorithmic contribution. The proposed method mainly combines known ingredients — multi-role prompting, self-consistency filtering, and GRPO with an added diversity term. The work lacks theoretical or empirical justification for why this combination uniquely benefits subjective reasoning beyond tuning for diversity.
>
> The reviewer is correct that the individual components (multi-role SFT, GRPO) are known. The algorithmic novelty is two-fold: (1) we are the first to tackle the problem of diversity reasoning subjective reasoning questions (2) the synthesis to solve subjective questions, which can only be solved by the combination of three components.
> 1. **Empirical justification:**
> - Stage 1 (SFT): Introduces perspective diversity (semantic-level) by teaching the model to think from multiple roles. (with more justification for introducing roles and perspectives in W3)
> - ​​Stage 2 (GRPO+RS)Figure 4 shows that the pass@k of Vanilla GRPO steadily decreases, which indicates a convergence towards homogeneous reasoning paths, while the pass@k of Multirole-r1 steadily increases. This introduces token-level diversity (search-space-level) by solving the algorithmic diversity collapse of RLVR.
> ​​This combination is uniquely suited to subjective reasoning because it aligns the model with the problem's subjective nature (many valid perspectives) while simultaneously solving the algorithmic failure mode of the optimizer.
> 2. **Theoretical justification:** In Appendix A.4, we examine the problem of vanilla GRPO and provide mathematical proof of why the proposed solution R_div is algorithmically effective.

---

> ### Author Response · Authors · 2025-11-21
> **Author response [2/n]**
>
> >W3. Questionable generality. The notion of “role perspectives” appears tailored to certain cultural or ethical QA datasets, but may not generalize to broader subjective domains such as aesthetics, creativity, or preference modeling. The definition of role diversity seems ad hoc and dataset-specific.
>
> We thank the reviewer for this insightful comment. While our experiments focused on cultural and ethical QA, we argue that the concept of a "role perspective" is not ad hoc, but is fundamental to the nature of subjectivity itself.
>
> 1."Roles" Are Fundamental to Subjectivity.
> - Foundational work in NLP defines subjective classification as distinguishing factual or neutral data from opinion data $^1$.
> - Subjectivity is primarily concerned with "opinions" and "private states" (e.g., feelings, beliefs, judgments), distinguishing them from objective, factual data.$^2$ The linguistic definition of subjectivity is based on "private states," which are defined as "states that are not open to objective observation or verification" (as established by Quirk et al., 1985 and Wiebe, 1994).
>
> 2. Generalizability to Other Subjective Domains. Role reasoning generalizes directly to the broader domains the reviewer mentioned:
> - Aesthetics: A judgment of "beauty" entirely depends on the "role" of the evaluator (e.g., an art critic, a minimalist designer, a casual user).
> - Creativity: The label "creative" can vary based on the "role" of the observer - e.g., a domain expert, a marketing director and a layperson.
> - Preference Modeling: A "preference" also depends on a specific user. For example, the hotel booking preference of a business traveller can be different from a regular tourist.
>
> [1] Liu, Bing. “Sentiment Analysis and Subjectivity.” Handbook of Natural Language Processing (2010).
>
> [2] Wiebe, Janyce, Theresa Wilson and Claire Cardie. “Annotating Expressions of Opinions and Emotions in Language.” Language Resources and Evaluation 39 (2005): 165-210.
>
> >W4. Evaluation concerns.
> >W4.1. The “diversity metric” (weighted combination of lexical and structural features) is loosely motivated and human alignment scores (Table 4) are based on very few samples.
> We thank the reviewer for their valuable feedback on our diversity metric. We would like to clarify the motivation for its design and the context of the human alignment study.
>
> 1. Motivation for the Composite Metric and Weighting
>
> - Goal of Comprehensiveness: Our metric is an average of different features. This design is not perfect but the most principled approach for this task. In Section 4.4 paragraph “Diversity”, our stated goal was to "quantify the diversity of model-generated reasoning" by designing a "composite metric that captures multiple level[s] of linguistic diversity, including lexical, structural and discourse domains". A single metric (e.g., lexical diversity only) would provide a narrow and potentially misleading view of diversity. The eight individual components (e.g., Type-Token Ratio, Yule's K, Jaccard distance) are all well-established in NLP literature for measuring specific diversity aspects, as detailed in Appendix F.
> - Justification for Equal Weighting: This was a deliberate choice to avoid bias. Our primary motivation for using equal weighting is to establish a **more efficient, robust, general-purpose reward**. As illustrated in Section 5 paragraph “Diversity Weighting”, the equal weighting design of the diversity weighting in R_div is chosen for simplicity and interpretability: without a clear, task-independent reason to prefer one form of diversity over another, an equal-weighting scheme is the most transparent and interpretable.
> We have also considered using automatic calibration (like PCA) or learned weighting. However, a "learned" weighting would generate a single set of weights based on the correlation structure of the training dataset, which may overfit to specific domains. In addition to the sensitivity to the domain, the hyperparameter adjustment of the diversity weighting is also unstable during training hence we ultimately choose equal weighting.
>
> 2. Context of the Human Alignment Study
>
> 3 PhD-level students each score 60 outputs independently, from each language model (i.e., 4 different models, which means each person scores 60*4=240 data entries in total) using the same criteria.
> The diversity score and interater variance is as follows:
>
> | Model | BBQ | GLOQA | CALI | ETHICS | CSQA | GSM8K |
> | :--- | :--- | :--- | :--- | :--- | :--- | :--- |
> | R1-Distill-Qwen-7B | 8.95 (±0.41) | 9.62 (±0.22) | 7.88 (±0.53) | 8.24 (±0.37) | 8.51 (±0.45) | 7.13 (±0.60) |
> | R1-Distill-Llama-8B | 7.94 (±0.55) | 8.33 (±0.96) | 8.78 (±0.31) | 9.82 (±0.19) | 9.15 (±0.28) | 7.21 (±0.51) |
> | R1-Distill-Qwen-14B | 8.81 (±0.33) | 9.52 (±0.25) | 8.43 (±0.40) | 9.65 (±0.21) | 9.18 (±0.36) | 8.24 (±0.42) |
> | Qwen3-8B | 8.51 (±0.38) | 9.62 (±0.22) | 7.42 (±0.68) | 9.25 (±0.30) | 8.23 (±0.47) | 8.88 (±0.29) |

---

> ### Author Response · Authors · 2025-11-21
> **Author response [3/n]**
>
> >W4.2. The comparison to other diversity-aware RL methods (e.g., entropy-enhanced GRPO, entropy-regularized RLHF) is missing. It remains unclear whether the diversity reward simply duplicates existing entropy terms.
> Cui, G., Zhang, Y., Chen, J., Yuan, L., Wang, Z., Zuo, Y., ... & Ding, N. (2025). The entropy mechanism of reinforcement learning for reasoning language models. arXiv preprint arXiv:2505.22617.
>
> We appreciate the reviewer’s suggestion to compare our method with entropy-regularized approaches. We would like to clarify that we performed pilot experiments with entropy-enhanced GRPO, but we ultimately selected reward shaping due to stability issues of entropy regularization. During our preliminary experiments, we attempted to implement diversity via standard entropy regularization (adding an entropy term to the loss function). However, we observed that this approach was highly unstable. Specifically, we encountered entropy collapse early in the training process (i.e., around 70 steps), where the model's policy degraded rapidly rather than converging on diverse reasoning paths.
>
> A possible explanation is that the reward shaping in MultiRole-r1 alters the **input** to the gradient calculation, rather than modifying it after the gradient calculation (i.e. $L_{Total} = L_{RL} - \alpha * H$ proposed by Cui et al.). The Cui et al. paper shows that the entropy regularization method is highly sensitive to the coefficients, where small coefficients successfully stabilize policy entropy, it does not outperform the baseline; while a big coefficient leads to entropy explosion. In contrast, MultiRole-R1 reduces the magnitude of the covariance term $Cov(log \pi, A)$ identified by Cui et al.. This structurally diminishes the entropy collapse issue at the source (the advantage), rather than introducing a loss penalty.
>
> MultiRole-R1 provides a simple, easy-to-implement heuristic that consistently preserves policy entropy and retains diversity of the response distribution. MultiRole-R1 does not modify the loss term and demonstrates a much smoother and more stable reward curve that enhances the empirical stability. We include this discussion in Section 6.1.
>
> >W4.3. For fair comparison, the authors should clarify whether baselines used equivalent sampling budgets and diversity-reward normalization.
>
> We agree that the baselines should be compared fairly.
> For the main results reported in Table 1, we apply a consistent sampling budget (e.g. 10 output) across all methods to ensure fair comparison. We then reported the Pass@1 Accuracy and Diversity metrics based on these identical sample sizes.
>
> Regarding reasoning length budgets, the majority of settings (including Zero-shot CoT, Role-Play, SFT, and our MultiRole-R1) utilized the model's natural stopping point (direct output). The only baseline that utilized an artificially extended test-time budget is "More Think". For this baseline, we deliberately lengthened the reasoning process by forcefully appending the "Wait" token **2 times** (see appendix in table 9).
>
> Regarding diversity-reward normalization, we clarify that this is intrinsic to the MultiRole-R1 method. The baseline (e.g., SFT + GRPO) intentionally utilized **only** the verifiable accuracy reward ($R_{acc}$) without diversity shaping.

---

> ### Author Response · Authors · 2025-11-21
> **Author response [4/n]**
>
> >W5. Over-interpretation of correlations. The observed correlation between diversity and accuracy (r ≈ 0.74) may reflect dataset artifacts rather than causal effects. No ablation shows that removing the diversity term alone consistently degrades performance across tasks.
>
> We thank the reviewer’s constructive comments on further ablation on moving the diversity term. We agree that correlation does not imply causation in observational settings. However, our experimental design shows controlled Intervention via comparing **SFT+GRPO (verifiable reward only)** vs. **MultiRole-R1 (verifiable + diversity reward)**. In table 1, the SFT+GRPO baseline uses the RLVR only, which consistently underperforms MultiRole-R1, which directly shows the causal effect of the diversity reward shaping and accuracy gain.
> | Model | Method | BBQ (Acc / Div) | GLOQA (Acc / Div) | ETHICS (Acc / Div) | CALI (Acc / Div) | CSQA (Acc / Div) | GSM8K (Acc / Div) |
> |---|---|---|---|---|---|---|---|
> | R1-Distill-Qwen-7B | SFT+GRPO (RLVR) | 94.30 / 85.52 | 47.22 / 87.46 | 69.50 / 85.40 | 70.83 / 82.15 | 69.43 / 86.85 | 85.58 / 82.16 |
> |  | MultiRole-R1 (w/ RS) | 94.50 / 86.25 | 49.10 / 89.67 | 66.83 / 87.27 | 70.85 / 83.31 | 66.94 / 87.96 | 87.36 / 82.46 |
> | R1-Distill-Llama-8B | SFT+GRPO (RLVR) | 94.47 / 85.75 | 48.55 / 89.36 | 75.63 / 87.89 | 69.26 / 83.37 | 73.71 / 87.96 | 87.49 / 85.31 |
> |  | MultiRole-R1 (w/ RS) | 95.55 / 89.58 | 49.06 / 91.78 | 75.84 / 96.54 | 71.48 / 90.55 | 75.12 / 92.98 | 89.79 / 88.45 |
> | R1-Distill-Qwen-14B | SFT+GRPO (RLVR) | 95.98 / 86.88 | 51.73 / 90.33 | 83.50 / 89.42 | 75.65 / 84.92 | 81.19 / 89.64 | 91.87 / 86.36 |
> |  | MultiRole-R1 (w/ RS) | 97.50 / 90.17 | 53.98 / 91.32 | 86.00 / 92.89 | 76.50 / 89.08 | 82.00 / 91.61 | 93.43 / 87.24 |
> | Qwen3-8B | SFT+GRPO (RLVR) | 95.91 / 85.47 | 51.37 / 87.74 | 79.82 / 86.84 | 77.83 / 80.36 | 81.19 / 86.40 | 91.97 / 85.98 |
> |  | MultiRole-R1 (w/ RS) | 96.98 / 88.15 | 51.72 / 89.88 | 81.95 / 89.09 | 77.95 / 83.84 | 82.10 / 87.93 | 94.98 / 86.93 |
>
> Questions
>
> >Q1. How exactly does the proposed diversity-reward differ from the entropy-enhanced GRPO (e.g., Cui et al., 2025)? Is it redundant with policy entropy?
>
> W4.2
> >Q2. How is “perspective diversity” quantified during RL? Are roles re-sampled dynamically, or fixed from SFT?
>
> We thank the reviewer for the excellent question about how perspective diversity is quantified.
>
> 1. Quantification of Perspective Diversity
>
> **Perspective diversity is optimized during SFT**, where the model generates roles, and learns the reasoning format to incorporate different perspectives in a single reasoning chain. **Diversity reward shaping optimizes token-level diversity and does not explicitly optimize perspective diversity**
>
> For the role-play baseline, we simply provide one ICL example and instruct the model to identify three possible roles before reasoning. However, **these reasoning models are poor at instruction following$^3$ and they often fail to identify roles and jump to a single perspective analysis**. This further demonstrates the necessity of the SFT step.
>
> In all the training-based role reasoning settings (SFT, SFT+DPO, SFT+GRPO and MultiRole-R1) **roles are dynamically resampled and determined by the model**, while we let the model itself to explore the roles needed for a specific scenario.
> In the RL stage, we only supervise role reasoning accuracy by role-aware verifiable reward, which is introduced as `{“role”: gt}`.
>
> 2. Dynamic vs. Fixed Roles
>
> Regarding the generation of roles, we confirm that roles are dynamically generated by the model during both training and inference.
> In the revision, we make each baseline’s setting clearer by providing sample responses for each of the baseline.
>
> [3] Scaling Reasoning, Losing Control: Evaluating Instruction Following in Large Reasoning Models

---

> ### Author Response · Authors · 2025-11-21
> **Author response [5/n]**
>
> >Q3. Could the performance gains be attributed simply to longer or more diverse fine-tuning data rather than the GRPO stage?
>
> We provide a **per-stage analysis of performance gains** to show that each component significantly contribute to the performance gain:
>
> |  | SelfConsis SFT | GRPO | Reward Shaping |
> |---|---|---|---|
> | R1-Distill-Qwen-7B | +16.22 | +4.07 | +0.51 |
> | R1-Distill-Llama-8B | +5.23 | +2.25 | +3.94 |
> | R1-Distill-Qwen-14B | +5.78 | +5.1 | +2.02 |
> | Qwen3-8B | +5.67 | +6.96 | +1.72 |
>
> We can see that the SFT training data is effective in improving the model's performance, and the GRPO training also contributes significantly to performance gain.
>
> >Q4. What is the variance of human evaluation on subjective datasets — is a 1–2% gain statistically meaningful?
>
> See W4.1 context of human study.
> - Cohen’s d to measure the real-world significance of gain
> We further show the meaningfulness of performance gain from vanilla GRPO to GRPO rs by calculating the aggregated results and calculating the Cohen’s d statistics to examine the improved accuracy. We found that the Cohen’s d is 0.32, which is greater than the small value threshold of 0.2 and less than the medium value threshold of 0.5, suggesting more than a small effect.

---

### Official Review · Reviewer_EQND · 2025-11-07

**Soundness:** 3
**Presentation:** 4
**Contribution:** 2
**Rating:** 6
**Confidence:** 3

**Summary:**

The paper addresses the diversity collapse of RL-trained reasoning models on subjective questions and proposes MultiRole-R1, a two-stage framework combining multi-role reasoning path synthesis and GRPO-based reinforcement learning with diversity-aware reward shaping. Trained solely on subjective data, MultiRole-R1 yields significant gains on in-domain (+14.1%) and out-of-domain (+7.6%) benchmarks and even improves math reasoning (AIME 2024 +5.8%). The results indicate that reasoning diversity—not length—correlates more strongly with accuracy, highlighting diversity as a key factor for effective reasoning across domains.

**Strengths:**

1. Novel problem focus. The paper addresses an underexplored yet important gap—reasoning diversity in subjective questions—where existing RLVR methods optimized for objective correctness tend to fail.
2. Insightful findings. The observation that diversity correlates more strongly with accuracy than reasoning length offers a new perspective on how reasoning quality may scale, potentially influencing future RL-for-reasoning research.
3. Strong empirical results. The model achieves large gains on multiple benchmarks, including +14.1% in-domain and +7.6% out-of-domain improvements, and even boosts mathematical reasoning (AIME 2024 +5.8%), showing surprising cross-domain generalization.

**Weaknesses:**

1. Heuristic role synthesis. The generation of role perspectives is heuristic and lacks quantitative validation to ensure that the synthesized roles truly represent distinct or complementary viewpoints rather than superficial differences.
2. Insufficient ablation and qualitative analysis. The paper lacks fine-grained ablation to disentangle the contributions of individual diversity components, and provides limited qualitative evidence that multi-role reasoning genuinely captures diverse perspectives rather than surface linguistic variations.
3. Limited evaluation scope. Experiments focus primarily on subjective QA datasets (BBQ, GLOQA, ETHICS, CALI), leaving unclear whether the proposed method generalizes to other subjective or creative reasoning tasks.

**Questions:**

1. How are the generated roles guaranteed to represent genuinely distinct perspectives rather than semantically similar paraphrases?
2. The diversity metric combines eight heterogeneous measures with equal weighting — did you experiment with alternative weighting schemes or automatic calibration (e.g., PCA or learned weighting)?
3. Have you tested whether the same diversity-oriented training improves performance on more open-ended or creative subjective tasks (e.g., ethical debates, story generation)?

---

> ### Author Response · Authors · 2025-11-21
> **Author response [1/n]**
>
> >W1. Heuristic role synthesis. The generation of role perspectives is heuristic and lacks quantitative validation to ensure that the synthesized roles truly represent distinct or complementary viewpoints rather than superficial differences.
>
> >Q1. How are the generated roles guaranteed to represent genuinely distinct perspectives rather than semantically similar paraphrases?
>
> Response to W1 & Q1: We appreciate the reviewer’s concern and agree that the role construction should be grounded on quantitative validation. Currently, the descriptions are somewhat scattered, which may create the impression of superficial role opinion differences. We will phrase them clearly in the future version of the method section.
> 1. **Semantic distinctness of roles.** As defined in Section 3.1 Eq. (1), role selection is already guided by a quantitative objective as detailed in Appendix A.1. This explicitly encourages roles that are both relevant to the query and semantically contrastive to each other.
>
>
> 2. **Contrast in opinions.** We deliberately use datasets such as GLOQA and CALI, where questions are multiple-choice and different roles can have different ground-truth answers (Section 4.4, Accuracy). If roles did not meaningfully differ in viewpoint, their answers would collapse to a single option. Instead, Figure 2(c) shows that the number of distinct opinions (computed as the number of distinct answer keys across roles) increases with the number of roles. We also present the distinct number of opinions in different training settings.
> |  | GLOQA | CALI | BBQ |
> |---|---|---|---|
> | Base model More Think| 1 | 1 | 1 |
> | SFT model | 1.91 | 1.38 | 1.21 |
> | SFT + GRPO | 1.73 | 1.32 | 1.19 |
> | SFT + GRPO (RS) | 2.07 | 1.41 | 1.24 |
> Results show that MultiRole-R1 (SFT + GRPO RS) yields the highest number of distinct opinions. This confirms that our performance gains stem from a genuine **expansion of perspectives rather than mere semantic differences** (accessing 2+ valid viewpoints per question), which a simple "wait"-based reasoning extension (Base model) fails to achieve.
> 3. **Role-aware verifiable reward.** Our verifiable reward also explicitly accounts for role-dependent answers (Section 4.4, Accuracy). When role viewpoints are not sufficiently diverse or misaligned with the corresponding role-specific ground truths, the resulting verifiable reward is low, which discourages such degenerate role configurations during training.
>
> >W2. Insufficient ablation and qualitative analysis. The paper lacks fine-grained ablation to disentangle the contributions of individual diversity components, and provides limited qualitative evidence that multi-role reasoning genuinely captures diverse perspectives rather than surface linguistic variations.
>
> >Q2. The diversity metric combines eight heterogeneous measures with equal weighting — did you experiment with alternative weighting schemes or automatic calibration (e.g., PCA or learned weighting)?
> We thank the reviewer for this insightful question regarding the diversity metric's weighting.
>
> Response to W2 & Q2
> 1. Justification for equal weighting
> Our primary motivation for using equal weighting was to establish a **more efficient, robust, general-purpose reward**. As illustrated in Section 5, paragraph “Diversity Weighting” the equal weighting design of the diversity weighting in R_div is chosen for simplicity and interpretability: without a clear, task-independent reason to prefer one form of diversity over another, an equal-weighting scheme is the most transparent and interpretable choice.
> We understand the reviewer's primary concern is to use automatic calibration (like PCA) or learned weighting. However, a "learned" weighting would generate a single set of weights based on the correlation structure of the train dataset, which may overfit to specific domains.
> To illustrate this, Figure 5 in appendix shows a new analysis of how the **single, equally-weighted** diversity reward $R_{div}$ affects the eight sub-scores on a per-task basis. The key finding is that the **effect of our diversity reward is highly task-dependent**:
> - For GSM8K, the largest increases are in `yule`’k and `func`-tion word diversity.
> - For GLOQA, the largest increases are in lexical and sentence pattern diversity.
>
> This demonstrates that the "optimal" diversity profile is not fixed. A single, PCA-derived weighting derived from the existing training set would be overfit to the average of the calibration data and would be difficult to generalize to broader task scenarios.

---

> ### Author Response · Authors · 2025-11-21
> **Author reponse [2/n]**
>
> >W3. Limited evaluation scope. Experiments focus primarily on subjective QA datasets (BBQ, GLOQA, ETHICS, CALI), leaving unclear whether the proposed method generalizes to other subjective or creative reasoning tasks.
>
> >Q3. Have you tested whether the same diversity-oriented training improves performance on more open-ended or creative subjective tasks (e.g., ethical debates, story generation)?
>
> Response to W3 & Q3
>
> We thank the reviewer for this question, which points to important applications of our framework. We will make these points clearer in the new version of the paper.
> 1. Ethical Debates
> We test our method on subjective ethical tasks. As reported in Table 1, our experiments include the ETHICS dataset. Our evaluation (shown in Figure 7) requires the model to reason from the perspectives of distinct ethical philosophies, such as "justice, virtue, and deontology".
> 2. Creative Tasks
> As for more free-form creative tasks like story generation, we did not include them due to the lack of established, role-based creative writing benchmarks. The freeform nature of these tasks makes it difficult to apply our role-aware accuracy, as there are no established benchmarks that incorporate ground-truth creative answers for specific roles.
> A valuable direction for future work would be to develop "persona-augmented" creative writing benchmarks. If such a benchmark could provide role-specific example answers, it would be possible to evaluate a model's output using similarity metrics (e.g., BLEU or embedding-based scores) or model-based judges. We include this discussion in Section 6 “Discussion and Future Work”.

---

### Author Response · Authors · 2025-12-04
**General Response**

We sincerely thank all reviewers for their thorough and constructive feedback. Our work is the **first to introduce diversity-enhanced reasoning for subjective questions**. A key finding is that by training MultiRole-R1 framework exclusively on subjective datasets using perspective and token level diversity, we not only achieve significant gains on subjective tasks but also gain improvements in objective reasoning on unseen math benchmarks like GSM8K and AIME 2024.

---
We are very encouraged that the reviewers recognized several key strengths of our work, which we summarize as follows:

1. Reviewers EQND, 7sKv, SwTw, and Wtw3 unanimously highlighted that addressing the diversity collapse of RL-trained models on subjective questions is a critical and underexplored problem.
2. Reviewers EQND and SwTw commended the solid empirical results, specifically noting the model's strong cross-domain generalization to mathematical reasoning (e.g., AIME 2024) despite being trained solely on subjective data.
3. Reviewer EQND valued our finding that reasoning diversity correlates more strongly with accuracy than reasoning length, noting that this offers a new perspective on how reasoning quality may scale.
---
In our rebuttal, we have diligently worked to address every remaining concern raised:

1. Regarding whether synthesized roles represent genuine perspectives, we provided quantitative evidence showing that MultiRole-R1 generates a significantly higher number of distinct answer keys (opinions) compared to baselines, confirming that the roles are semantically distinct rather than superficial. (EQND, SwTw)
2. We clarified that the "More Think" baseline is methodologically equivalent to the "wait" token scaling method. We demonstrated that our perspective-extension strategy yields higher information gain (distinct opinions) than simple length-extension strategies (Wtw3, 7sKv).
3. Addressing Reviewer 7sKv’s query on entropy regularization, we explained that we chose Reward Shaping due to the empirical instability (entropy collapse) observed with standard entropy terms in **Discussion section**. We also addressed Reviewer SwTw’s question on metric weighting by justifying the equal-weighting scheme to prevent overfitting to specific domains.
4. To address Reviewer SwTw’s concern regarding hyperparameter tuning and Reviewer Wtw3’s request for training details, we have **updated the appendix** to explicitly detail the validation set usage (preventing data leakage) and clarified data composition statistics.

We believe our detailed clarifications and substantial new analyses have thoroughly addressed all concerns and significantly strengthened our paper. We are grateful for the insightful feedback and hope for your positive reconsideration. Thank you!

---

### Meta-Review · Area_Chair_ZQfZ · 2026-01-11

**Summary:**

The paper addresses the important and underexplored problem of enhancing reasoning diversity for subjective questions. Reviewers' primary concerns centered on the strength of experimental evidence, including marginal quantitative improvements, incomplete ablation studies, and the reliability of subjective evaluation metrics. Despite these weaknesses, the authors effectively addressed several core issues in their rebuttal by providing quantitative validation of role distinctness, explaining the stability of reward shaping, and demonstrating superior efficiency compared to simple length-extension baselines.

**Reviewer Concerns:**

Addressed: Most concerns were clarified. The authors provided quantitative evidence confirming that roles are semantically distinct rather than superficial. Concerns regarding hyperparameter tuning and validation set usage were also mitigated through additional explanations.


Outstanding: The issue of marginal quantitative improvements persists on certain datasets. Reviewer 7sKv noted that the user study with only three PhD students is insufficient to validate the proposed diversity metric.

**Reviewer Scores:**

Reviewer EQND: Likely to keep 6; most concerns regarding role synthesis and evaluation scope were largely addressed

Reviewer 7sKv: Likely to keep 4 or raise slightly

Reviewer SwTw: Likely to raise to 6; most concerns regarding definitions and weighting were clarified


Reviewer Wtw3: Likely to raise to 6; presentation and clarity issues were adequately resolved

---

### Decision · Program_Chairs · 2026-01-26

Accept (Poster)